

# Grounding line migration through the calving season of Jakobshavn Isbræ, Greenland, observed with terrestrial radar interferometry

Surui Xie[1], Timothy H. Dixon[1], Denis Voytenko[2], Fanghui Deng[1], and David M. Holland[2,3]

[1]School of Geosciences, University of South Florida, Tampa, FL, USA
[2]Courant Institute of Mathematical Sciences, New York University, New York, NY, USA
[3]Center for Global Sea Level Change, New York University, Abu Dhabi, UAE

*Correspondence to:* Surui Xie (suruixie@mail.usf.edu)

**Abstract.**

Ice velocity variations near the terminus of Jakobshavn Isbræ, Greenland were observed with a terrestrial radar interferometer (TRI) during three summer campaigns in 2012, 2015, and 2016. Ice velocity variations appear to be largely modulated by ocean tides. We estimate a ∼1 km wide floating zone near the calving front in early summer of 2015 and 2016, where ice moves in

phase with ocean tides. Digital Elevation Models (DEMs) generated by the TRI show that the glacier front here is thin (ice surface is <125 m above local water level). However, in late summer 2012, there is no evidence of a floating ice tongue in the TRI observations. Ice surface elevation near the glacier front was also higher, >140 m above local sea level within a very short distance (<1 km) from the ice cliff. We hypothesize that during Jakobshavn Isbræ's recent calving seasons, the ice front advances ∼3 km from winter to spring, forming a >1 km floating ice tongue. During the subsequent calving season in mid- and

late-summer, the glacier retreats by losing its floating portion through a sequence of iceberg calving events. By late summer, the entire glacier is likely grounded. In addition to ice velocity variations driven by tide rise and fall, we also observed a transverse velocity variation in the mélange and floating ice front. This across flow-line signal is in phase with the first time derivative of tidal height, and is likely associated with tidal currents or bed topography.

## 1 Introduction

Greenland's largest marine-terminating glacier, Jakobshavn Isbræ, has doubled in speed and retreated tens of km in the last few decades (Joughin et al., 2004; Rignot and Kanagaratnam, 2006; Joughin et al., 2008; Howat et al., 2011). This process has been attributed to several processes, including increased subsurface melting and iceberg calving triggered by relatively warm ocean water (Holland et al., 2008; Motyka et al., 2011; Enderlin and Howat, 2013; Myers and Ribergaard, 2013; Truffer and Motyka, 2016). In recent years, the glacier has maintained a relatively stable terminus position despite continued speedup, primarily due

to the fact that the glacier is now embedded in the ice sheet, with large inflows of ice from the sides supplying ice to the main glacier channel, albeit with some thinning (Joughin et al., 2008). However, it is not clear if the position and status are stable, as Jakobshavn Isbræ has a down-dipping upstream bed (Clarke and Echelmeyer, 1996; Gogineni et al., 2014). Some numerical models suggest that glaciers with reverse bed slopes cannot maintain stable grounding lines, as bed topography favors ingress of warm fjord bottom water, accelerating melting at the ice-ocean interface (e.g., Vieli et al., 2001; Schoof, 2007).



In addition to the dramatic secular speedup and retreat, there are strong seasonal variations in both ice speed and front position at Jakobshavn Isbræ. These have a strong inverse correlation: ice speed accelerates through spring and summer but slows down in winter, while glacier front position retreats from spring to summer, reaching a minimum in late summer when ice speed is maximum (Joughin et al., 2008). This supports the hypothesis that loss of the buttressing ice tongue during the calving
season contributes to Jakobshavn Isbræ's seasonal speedup. The rapid acceleration since 2000 may thus be the sequential result of losing its large floating ice tongue from 1998 to 2003 (Joughin et al., 2004, 2008). By investigating interactions between the glacier and its proglacial ice mélange, Amundson et al. (2010) interpreted the seasonal advance and retreat of the glacier terminus as an effect of seasonally-variable rheology in the ice mélange: stiffened mélange in winter suppresses major calving events, enabling the terminus to move forward; while in summer, a weaker mélange can no longer prevent major iceberg
calving, and the terminus retreats. They used a force balance analysis to demonstrate that large-scale (full-glacier-thickness icebergs) calving events are not likely to occur when the ice front is well-grounded. Based on this, they suggested that one of the necessary conditions for frequent full-glacier-thickness iceberg calving at Jakobshavn Isbræ is a floating or close-to-floating terminus in summer.

Currently, there is no efficient approach to observe basal conditions of marine-terminating glaciers directly. However, some
basal conditions, such as grounding line position, can be inferred from measured ice motion (Heinert Riedel, 2007; Rignot et al., 2011; Rosenau et al., 2013). It has been observed that for many marine-terminating glaciers, ice speed is affected by ocean tides (e.g., Makinson et al., 2012; Podrasky et al., 2014; Voytenko et al., 2015a). At Jakobshavn Isbræ, Podrasky et al. (2014) used GPS and theodolite data obtained in a two-week campaign in middle to late August 2009 to study velocity response to ocean tidal forcing near the terminus of Jakobshavn Isbræ. After removal of large background speed and perturbation caused
by a single calving event, tidal forcing explained 10%–90% of the remaining signal. Based on the fast decay of tidal responses upstream, they concluded that the terminus region is very nearly grounded during summer months. Rosenau et al. (2013) used photogrammetric time-lapse imagery to estimate groundling line migration and calving dynamics at Jakobshavn Isbræ. They found that the groundling line retreated 3.5 km from 2004 to 2010, with an ephemeral floating tongue during the advance season.

In this study, we use ice velocity and elevation time series observed with terrestrial radar interferometry (TRI) to analyze groundling line position and tidally affected ice flow. Previous work (Peters et al., 2015; Voytenko et al., 2015a, b, c, 2017; Holland et al., 2016; Xie et al., 2016) has shown that TRI can overcome the limitations of GPS (low spatial resolution, difficult to deploy near the calving front), theodolite (low spatial resolution and precision), photogrammetry (low reliability in bad weather and at night), and satellite observations (low temporal resolution). Here we use TRI measurements obtained in three
summer campaigns to investigate tidal responses and the evolving glacier front through a calving season.

## 2   Data Acquisition

We observed the terminus of Jakobshavn Isbræ in three summer campaigns in 2012, 2015, and 2016. Each campaign obtained a continuous record of velocity and elevation change over 4 to 13 days. The TRI instrument is a real-aperture radar operating





at Ku-band (1.74 cm wavelength) and is sensitive to line-of-sight (LOS) displacements of ∼1 mm (Werner et al., 2008). It has one transmitting and two receiving antennas, which allows for high spatial and temporal resolution measurements of both displacement and topography. The antennas are rigidly attached with a rack structure, which sits on a motor that rotates around a fixed vertical axis. In 2012, the instrument was deployed on a tripod reinforced with sandbags, with the calving front

∼3–6 km away. In 2015 and 2016, the instrument was mounted on a metal pedestal that was bolted ∼10 cm deep into solid rock, and protected by a radome to eliminate disturbance from wind and rain, with the calving front ∼2–5 km away. In all three campaigns, the radar scanned to a maximum distance of 16.9 km, generating images with both phase and intensity information. The resolution of the range measurement is ∼1 m. The azimuth resolution varies linearly with distance, and is determined by the arc length $l = D{\cdot}A$, where D is the distance to the radar, and A is the azimuth angle step in radians. In all three campaigns,

the azimuth angle steps were set to be 0.2°, resulting in an azimuth resolution of 7 m at 2 km distance, 14 m at 4 km, etc.. Other parameters in these measurements are listed in Table 1. Figure 1 shows the spatial coverage of measurements in each campaign.

## 3  Data analysis

### 3.1  TRI data processing

TRI data were processed following Voytenko et al. (2015b): 1) slant range complex images were multi-looked to reduce noise; 2) interferograms were generated between adjacent scans; 3) A stationary point on rock was chosen as reference for phase unwrapping. Unwrapped phases were then converted to line-of-sight (LOS) velocities. We define LOS velocity as positive when ice moves towards the radar, and negative when ice moves away from the radar. All results were resampled into 10 m × 10 m pixel spacing maps unless otherwise specified, with a bicubic spline interpolation algorithm. To georeference the

TRI results, we used a Landsat-7/8 image acquired during (if not, <2 day time difference) the observation period as reference. By fixing the radar location and horizontally rotating the intensity image, a rotation angle was estimated based on the best match of distinct surface features (e.g., coast line, ice cliff, icebergs, etc.), thus TRI-derived results were georeferenced into the earth reference system. In this study, we used the polar stereographic projection to minimize distortion. Notice that the TRI instrument measures LOS intensity and phase information. Converting LOS data into x-y grid coordinates induces some

distortions due to topography, especially in the mélange close to the radar, where height difference is the largest. The radar location in 2012 was ∼280 m above local water level, and in 2015/2016 ∼200 m above local water level. A simple calculation based on geometry shows that distortion due to topography is <15 m. There are two other error sources in georeferencing TRI data: 1) Radar position error (it was measured with a single frequency GPS, with location error estimated at less than 10 m); 2) Rotation error in matching TRI and Landsat images. By comparing georeferenced TRI images with different Landsat-7/8

images, we found no visible mismatch larger than 4 pixel width of the satellite images. We thus assess that the coordinate error in georeferenced TRI results is <60 m, i.e., smaller than 4 pixel (typically <2 pixel) width of Landsat-7/8 panchromatic images. Moreover, because the radar was deployed on a fixed point for each respective campaign, and we used the same radar coordinates and rotation angle in georeferencing for each campaign, the error due to georeferencing will not affect time series





analysis. Other errors in TRI data, such as phase variations associated with variable atmospheric water vapor, are difficult to model. To minimize water vapor effects, we only analyzed data within 10 km of the radar unless otherwise specified.

TRI data obtained in 2015 have been previously discussed in Xie et al. (2016). The same data are used here, but we added 17 h of additional data obtained before the period analyzed by Xie et al. (2016). The additional data were acquired when the instrument was in an experimental mode: rather than 150°, the scanned arc was sometimes set to different values, and the repeat time was sometimes 1 or 2 min rather than 1.5 min. Otherwise, the additional data have the same quality as the subsequent acquisitions. We processed the additional data with the same standards and converted it into the same reference frame as the remaining 2015 data.

Except for several rapid changes in velocity caused by calving events, the processed results from 2015 and 2016 show good continuity. However, velocities from 2012 have some significant offsets (Supplementary Fig. S1(a)). Most of these offsets reflect phase unwrapping errors, reflecting an incorrect integer multiples of microwave cycles has been applied during the phase unwrapping process. The repeat time in 2012 (3 minutes) was longer than the other two years, and ice motion relative to the adjacent areas in the radar LOS during that interval could exceed 1 radar wavelength. We fixed the phase offsets in 3 steps: 1) Estimating the velocity time series at a single point on the ice; 2) Using this kinematic point as the reference point for phase unwrapping to get relative velocities for all other points mapped; 3) Adding the velocity model from step 1 to the relative velocities. We compared this new velocity map with velocities estimated by feature tracking, which is independent of interferometry and does not require phase connection. The phase jumps are greatly reduced, and we believe the resulting velocity time series are an accurate indicator of ice motion. Details are given in the Supplementary Text S1.

## 3.2 Tidally driven ice motion analysis

The calving front is where the glacier directly interacts with the ocean. By changing back-pressure on this front, ocean tides are known to influence the behavior of some marine-terminating glaciers (Walters, 1989; Anandakrishnan and Alley, 1997; Podrasky et al., 2014). Besides back-pressure, a full-Stokes nonlinear viscoelastic model (Rosier and Gudmundsson, 2016) suggests that when there is a floating ice tongue, tidal flexural stress can also be an important forcing for marine-terminating glaciers. In addition, tidal variation can influence basal friction at the ice-bed interface, thus changing the sliding rate of the glacier (e.g., Walker et al., 2013; Voytenko et al., 2015a).

For all three campaigns, velocities near the terminus show significant semi-diurnal variations, and perhaps a small diurnal signal. Figure 2 shows the power spectral density analysis (PSD) for selected data in 2016. PSDs for 2012 and 2015 are shown in the Supplementary Fig. S6 and S7. Previous studies indicate that apart from calving events, short-term ice velocity variations at Jakobshavn Isbræ are well described with simple tidal response models (e.g., Rosenau et al., 2013; Podrasky et al., 2014). Diurnal variation caused by surface melting may also contribute to velocity variation. This has been observed at both Jakobshavn Isbræ (Podrasky et al., 2012) and Helheim Glacier (Davis et al., 2014)). Due to the short time span of our data, it is not possible to recover the full temporal spectrum of ice velocity variations. Instead, we focus on the largest spectral components of the velocity field.



There was no tide record in the fjord near the terminus during our campaigns. Podrasky et al. (2014) analyzed a 14 day tide record in the fjord within 5 km of the calving front obtained in August 2009, and compared it with a longer record from Ilulissat. The two datasets show close agreement, with no measurable delay in time, and a maximum difference in stage <10 cm. Thus they used the longer record of tides at Ilulissat to analyze the tidal response of the glacier. Similarly, we used analyzed

tidal constituents from the long-term record at Ilulissat to predict the tides in the fjord during our campaigns. Richter et al. (2011) applied harmonic tidal analysis to 5 years of long-term sea-level records at Ilulissat and estimated that the largest 3 tidal constituents are K1, M2 and S2, with amplitudes of 0.331 m, 0.671 m and 0.273 m respectively. These three constituents account for >95% of all the tidal constituents analyzed by Richter et al. (2011). Figure 3 shows the predicted tide and tidal rate (defined as the $1$st time derivative of tidal height) during the 2015 campaign, when we had a mooring deployed at the mouth

of the fjord (red hexagon in Fig. 1) that recorded tidal height. There are only small differences between measured tide or tidal rate with predictions using the three largest constituents. In the following analysis, we focused on ice velocities with the same frequencies as the K1, M2 and S2 tide constituents. Other components of motion with similar frequencies will be aligned into these 3 constituents. For example, diurnal variation caused by surface melting with a period of ∼1 d, if exists, will not be separable from K1 with a period of 1.0027 d.

Many tidal response models analyze the response of ice position to tidal height variation (e.g., Davis et al., 2014; Podrasky et al., 2014). However, our TRI measurement is only sensitive to LOS displacement. The corresponding velocity derived by interferometry is the $1$st time derivative of LOS displacement. Velocity can be converted to position by integration, however, due to data gaps and the nonlinear behavior of the velocity time series, integration of velocity time series may introduce artifacts. Therefore, we used ice velocity instead of position and analyzed the response of ice velocity to tidal rate. The amplitude will

be amplified by frequency (signal with a higher frequency will have a larger range of $1$st time derivative, see Supplementary Text S3), but the phase difference is unchanged by differentiation.

Before the tidal response analysis, we used the modified Z-score method (Iglewicz and Hoaglin, 1993, also see Text S1 in the Supplement) to remove outliers. We note that TRI-observed ice motion in the mélange is very sensitive to even very small calving events, while ice on the glacier is less sensitive to small calving events. For the 2012 data, due to frequent calving

events, we were not able to phase unwrap the full time series. Instead, we used data obtained from 6 August to 10 August when there was only one small calving event (see Supplementary Fig. S1) for the following analysis. For the 2015 data, there were many small calving events and a large one at the end (Xie et al., 2016), which resulted in a noisy time series for the mélange. We therefore omitted the 2015 mélange from further analysis. For 2016, a step-change in ice elevation (dashed purple line in Fig. 2, also marked by red arrow in Fig. 4) was observed, separating the mélange into two distinct parts. Downstream from the

step-change, ice motion is very noisy and difficult to analyze for periodic signal. Upstream from that, ice velocity variation is similar to the glacier. Therefore, we did not do tidal response analysis for the ice mélange downstream from the step-change zone in 2016. Movie S1, S2, and S3 in the Supplement show all major calving events observed during the three campaigns, and consequent changes in the mélange.

A comprehensive study of the step-change in the 2016 mélange is beyond the scope of this paper, but we note that many

calving-like events occurred near this front. Ice surface height upstream of the step-change is ∼10 m higher than downstream



from the step-change (see Supplementary Text S4 and Fig. S8(c)). We hypothesize that this mélange feature is a consequence of rapid calving which produces tightly packed ice, whose surface is strongly affected by fjord geometry. Three types of geometric features may be responsible: 1) the ice mélange has arched as the flow channel adjacent to the calving front is relatively narrow, limiting escape of calved ice and contributing to high compressive stress. The step change in topography is analogous to a thrust

fault in bedrock subject to high compressive stresses.. 2) Bedrock-driven ice flow direction changes: ice immediately adjacent to the calving front flows along a southeast-northwest direction, while a few km downstream from the cliff, ice flow changes to a nearly east-west direction. Bedrock may be a barrier for new calved ice changing its flow direction, thereby creating a crest. 3) An underwater moraine could limit the motion of large icebergs, compressing near-field ice and contributing to the step-change. Decadal changes in the calving front position (Fig. 4) show that the location of the step-change coincides with

the glacier terminal positions for several years since 2005. This, along with debris left by historic glacier activities, could form a moraine within the fjord. While available bed bathymetry models for Jakobshavn Isbræ do not show such a feature, they have limited resolution, and small topographic features could be missed. The latest bed elevation model (An et al., 2017), is based on inversion of helicopter-borne gravity and radar depth sounder data, with a few depth points measured by eXpendable Current profiler (XCP) and Conductivity Temperature Depth (XCTD) probes (4 lay within the gravity survey). This model has

a spatial resolution of 750 m, and an average height precision of ∼60 m. Future measurements around the step-change would help to identify relevant topographic features. Additional analysis on the mélange step-change observed in the 2016 TRI data can be found in the Supplementary Text S4.

For both 2012 and 2015 campaigns, ∼4 day data were analyzed and a *2nd-order* polynomial was used to detrend the time series. For the 2016 campaign, ∼13 days of data were analyzed. This time series shows significant responses to a few calving

events (Fig. 5). We used a function composed of a 2nd-order polynomial + 3 pairs of sines and cosines to estimate the response to calving events, and then removed the polynomial. The function is:

$$V_i = a_j + b_j t + c_j t_i^2 + \sum_{k=1}^{n} [d_k \sin(2\pi f_k t_i) + e_k \cos(2\pi f_k t_i)] \tag{1}$$

where $V_i$ is the observed LOS velocity at time $t_i$. $a_j$, $b_j$, and $c_j$ are coefficients of *2nd-order* polynomial for the $j$th period (n$j$ in total), periods are separated by noticed calving events. To better estimate the *2nd-order* polynomial, periods shorter than 1

25 day are not used. $d_k$ and $e_k$ are coefficients of the $k$th periodic component, with frequency $f_k$ among those of K1/M2/S2 tide constituents. After this, response to calving events and tidal constituents with periods >2 day are largely eliminated. Figure 5 gives an example of the observed and detrended time series. Note that data in 2016 span longer times than 2012 and 2015. To save computational time, we converted TRI images into pixel sizes of 30 m × 30 m for a map-wide analysis.

Detrended time series were passed through a median filter to reduce noise. The kernel size is 3/5/5 for data in 2012/2015/2016,

equal to a 9/7.5/10 minute time window. After that, all time series were analyzed using the method of Davis et al. (2014), which estimates the amplitudes and phases of the three periodic components with the same frequencies as the K1, M2 and S2 tidal constituents. This method allows us to distinguish components with close frequencies (in our case, M2 and S2). We also used





a least squares fit to an equation with 3-frequencies sine/cosine as an alternative method. The two methods fit the time series equally well, with differences that are insignificant compared to noise.

Figure 6(b,d,f) shows maps of phase lag (converted to time in h) from tidal rate to TRI observed LOS velocity at the M2 tidal frequency, along with a velocity profile for each campaign. Note that due to the phase character of periodic signals, dark

red on the map represents phase values that are close to dark blue. For example, 12.42 h (period of M2) "equal" to 0. Note also that the phase lag maps only show pixels with signal-noise-ratio (SNR) > 1.5, where we define SNR as:

$$SNR = \frac{\sigma_{\mathrm{signal}}^2}{\sigma_{\mathrm{noise}}^2} \qquad\qquad (2)$$

and use the root-mean-square (RMS) of the velocity time series to represent $\sigma_{\mathrm{signal}}$, and RMS of the residuals to represent $\sigma_{\mathrm{noise}}$. We use the M2 tidal signal to illustrate tidal responses in this paper since this is the largest tidal constituent. Phase lag maps

for K1 and S2 are shown in the Supplementary Fig. S10, with patterns that are similar to M2.

Figure 6 shows two types of phase lag patterns. For 2012, LOS velocity of ice in the mélange has ∼0 phase lag to tidal rate, whereas the phase lag increases sharply at the ice cliff, to ∼8.5 h on the glacier front. For both 2015 and 2016, there is a narrow zone at the glacier front that is in phase with the tidal rate, with phase lag close to 0. Upstream from that, phase lag increases to ∼8 h.

**4  Discussion**

### 4.1  Grounding line variation in a calving season

One hypothesis concerning the annual cycle of advance and retreat of Jakobshavn Isbræ is that a floating tongue grows in winter and disappears in late summer (Joughin et al., 2008; Amundson et al., 2010). However, there are no direct observations through a full calving season. We addressed this by assuming consistent behavior over the five year observation period, and

considering our data to be a representative sample of early and late melt season behavior.

Rosenau et al. (2013) looked at the cross correlation coefficient between tidal height and the vertical component of ice trajectory to estimate grounding line migration. This approach assumes that the only force that drives vertical ice motion is the change of buoyancy due to tide rise and fall. From an analysis of optical images, they found no evidence of floatation in mid-July 2007 (∼6 day duration), a ∼500 m wide floating zone from 8 August to 9 August 2004 (∼1 day duration), and an

even wider floating zone from late spring to early summer 2010 (∼29 day duration). Podrasky et al. (2014) applied a tidal admittance model to analyze both horizontal and vertical responses to tidal forcing at Jakobshavn Isbræ. They found rapid decay of admittances at the glacier front, corresponding to small (∼2 km and ∼0.7 km for horizontal and vertical, respectively) *e*-folding lengths (the distance over which the amplitude decreases by a factor of *e*), from which they concluded that the glacier front was very nearly grounded in late August 2009.

TRI-derived LOS velocities reflect the influence of several forcing processes. Surface meltwater-induced velocity variation is a quasi-diurnal signal. Podrasky et al. (2012) detected an amplitude of up to 0.1 m d$^{-1}$ diurnal signal 20–50 km upstream from





the terminus of Jakobshavn Isbræ. The timing of the diurnal maxima was ∼6 hours after local noon, which corresponds with surface melting. Within 4 km to the ice cliff, Podrasky et al. (2014) found diurnal variations that are 0.5–1 times the amplitude of tidally-forced variations, with a maxima 10.9–11.7 hours after local noon. At Helheim Glacier, Davis et al. (2014) identified a signal with peak-to-peak variation of ∼0.7 m d$^{-1}$ in glacier flow speed at a site close to the terminus, likely associated with

changes in bed lubrication due to surface melting. While surface meltwater can cause a diurnal component in ice velocity, it should have no direct influence on semi-diurnal signals, which are the dominant signals observed in all three of our campaigns. Supraglacial lakes were not observed near the terminus during our campaigns. Upstream from the terminus, supraglacial lake drainage events occur but are sporadic. Podrasky et al. (2012) observed at most three supraglacial lake drainage events near the terminus during 3 summers from 2006 to 2008. If such events occurred during our data collection periods, the responses are

likely to have been eliminated by the detrending process.

The LOS velocity variation contains two components of ice motion: 1) vertical motion; 2) horizontal motion. For all three campaigns, the radar was always located higher than the ice surface in the mélange and the first ∼3 km of the glacier. In this case, the TRI-observed LOS velocity vertical component is:

$$V_{los} = \frac{1}{\sqrt{(\frac{L}{H_0 - h})^2 + 1}} \frac{dh}{dt} \tag{3}$$

where $L$ is the horizontal distance between radar and the target, $H_0$ is the mean height of the target, $h$ is the vertical movement relative to $H_0$, and $\frac{dh}{dt}$ is the vertical component of ice velocity. We presume that for floating ice, $\frac{dh}{dt}$ is correlated with the tidal rate. Hence, in the mélange, $\frac{dh}{dt} \approx$ tidal rate; on the glacier, $\frac{dh}{dt}$ will have a smaller range compared to tidal rate due to energy decay of tidal flexure, but the ratio can be close to 1 if ice near the cliff is very weak, similar to what Voytenko et al. (2015a) found at the terminus of Helheim Glacier. For grounded ice, $\frac{dh}{dt}$ variation should have much smaller amplitude

compared to tidal rate variation. Horizontally, for all three campaigns, ice on almost the entire glacier moves towards the radar (LOS velocity is positive, see Supplementary Fig. S3, S4 and S5). Previous studies suggest that several mechanisms are acting simultaneously, and there is no defined phase relation between tide variation and ice speed (e.g., Thomas, 2007; Aðalgeirsdóttir et al., 2008; Davis et al., 2014; Podrasky et al., 2014). However, at the terminus of Jakobshavn Isbræ, Podrasky et al. (2014) found that glacier speed and tidal height are anti-correlated. This likely reflects variation of back-pressure forcing associated

with tide rise and fall.

We have not attempted to derive a comprehensive model for ice velocity variation caused by changes of back-pressure or other factors. Instead, we adopt the admittance parameters estimated by Podrasky et al. (2014) to assess a near-upper bound of along flow-line velocity variation. Using theodolite and GPS observations near the ice front, Podrasky et al. (2014) estimated horizontal and vertical tidal admittances of <0.12 and <0.15, respectively. In terms of phase, tide-induced vertical motion is

in phase with the ocean tide, while horizontal velocity is anti-correlated with tidal height, i.e., horizontal velocity maxima are concurrent with the inflection points of tidal rate. By assuming the glacier was under the same conditions as the time when Podrasky et al. (2014) did their measurements, we predict ice velocities near the glacier front. In Fig. 7, *F1* and *F2* correspond to the two points marked with purple triangles in Fig. 6(f). For each point, two components of ice velocity were predicted



and projected onto the LOS direction to the radar: 1) vertical velocity by using tidal admittance of 0.15, and time lag of 0 to tidal rate, shown by solid black curve; 2) horizontal velocity by using tidal admittance of 0.12, and anti-correlated with tidal height, shown by the dashed black curve. The red curve shows the sum of these two components. Podrasky et al. (2014) inferred that the glacier front was very near grounded during their observation period, and both horizontal and vertical tidal

admittances dropped dramatically upstream. While we use the upper bound of the tidal admittance by Podrasky et al. (2014), the amplitudes of our predicted velocities are almost the maxima for grounded ice. However, as shown in Fig. 7, predicted tide-induced vertical velocities have far smaller magnitude than our TRI-derived velocities – the horizontal component is larger, but is negatively-correlated with TRI observations. Therefore, we reject the hypothesis that ice near the cliff in 2016 was near grounded as during the observation period of Podrasky et al. (2014) in late August. For comparison, we also plot predicted

LOS velocities by assuming ice was in a free flotation state, shown in blue. This is in-phase with the TRI derived velocities, although the magnitude does not fully explain the larger signals observed by TRI. Possible reasons are discussed below.

Ice located in the low phase lag zone (dark red or blue in Fig. 6(d)) in 2015 yields similar results. For ice further upstream in 2015 and 2016, and almost the entire glacier front of 2012, we can not reject the possibility of a near-grounded basal condition, because the admittances by Podrasky et al. (2014) can then produce LOS velocities that are sufficient large and correlated with

TRI observations. Figure 8 shows predicted (red curve) and observed velocity (grey dots) of a surface point (*B1* in Fig. 6(b)) that is immediately adjacent to the cliff during our 2012 campaign. These two have similar amplitude and phase, though the maxima of TRI-observed velocity are not explicitly concurrent with the inflection points of tidal rate, instead, they are slightly earlier ( 0.5 h) than the inflection points of tidal rate. We presume that ice in the high phase lag zone in Fig. 6 is either grounded, or nearly grounded.

Based on our analysis, we hypothesize that during early summer 2015 and 2016, there was a narrow zone of floating ice near the cliff, which is at least the width of the low phase lag zone (∼1 km). However, we are unable to determine if ice more than 1 km from the cliff is grounded or not. The annual maximum and minimum extents of the ice front (solid/dashed pink lines in Fig. 6) support our hypothesis: the low phase lag zone on the glacier during both the 2015 and 2016 observations coincides with the transition zone between maximum and minimum glacier front. In contrast, for the 2012 data, the ice cliff was close

to the annual minimum. Additional evidence to support this hypothesis comes from the ice surface elevation map. Figure 9 shows the median average DEM from a 1 day of TRI measurements for each campaign. In 2012, near the centre-line of the main trough, surface ice elevation increases dramatically near the glacier front, to >140 m in <1 km distance from the cliff. In contrast, in 2015 and 2016, ice elevation increases more slowly, with a ∼1 km wide zone that is <125 m higher than local sea level. In the low elevation zone, the overall buoyancy could make the condition favorable for a floating glacier front.

Figure 10 shows the annual maximum and minimum extent of Jakobshavn Isbræ from 2012 to 2016. During the time span of our TRI campaigns, the glacier front maintained a relatively consistent position, with ∼3 km ice advance and retreat per year. Time series of satellite images also suggest that in late summer to early autumn, the glacier front usually stabilizes near the minimum position for a few weeks before a steady advance. Using the TRI campaign in 2012 as a proxy for late summer conditions, and campaigns in 2015 and 2016 as proxies for early summer conditions, we infer that from 2012 to 2016,

Jakobshavn Isbræ had a floating tongue in the early stage of the calving season. Under-cutting and tidal flexure then weakened

the floating ice, leading to large calving events in subsequent months. During the calving season, calved ice surpassed ice flow into the terminus zone, causing the glacier front to retreat. In late stages of the calving season, the glacier had lost the majority of its floating portion, and the ice front became grounded or nearly grounded. Figure 11 depicts the glacier front in early and later summer. In early summer, ice near the calving front moves in phase with tides (shown by double sides arrow in

Figure 11a).

## 4.2  Other sources of forcing

Figure 7 shows that even by assuming ice is free-floating near the cliff, LOS velocity variation generated by tide rise and fall is insufficient to explain the observed velocity time series. Ice velocity variation caused by surface melting, if in phase with tidal rate, can increase the overall velocity variation. In this study, we did not separate the quasi-diurnal signal associated with

surface melting from similar tidal components. However, there is some evidence of such a signal. As shown in Fig. 12, in the low phase lag zone of 2016, if we assume diurnal along-flow-line speed variation (frequency = 1 cpd) with maxima a few h after local noon, e.g., 6 h after local noon, the related LOS velocity observed by TRI would be either correlated (when ice moves towards the radar, shown in dashed red curve) or anti-correlated (when ice moves away from the radar, shown in dashed blue curve) with LOS velocity caused by tide rise and fall. This may partly explain why the normalized PSD in Fig. 2(C) has less

obvious diurnal constituents compared to Fig. 2(A,B): due to the geometry difference, TRI-observed LOS velocity is negative in box C, but positive in box A and B. The diurnal tide signal will be superimposed on a negative or positive diurnal signal associated with surface melting, decreasing or enhancing the observed signal. However, surface melting should not make a significant contribution to semi-diurnal signals, as it is a quasi-diurnal phenomena. In addition, most sources of forcing would induce longitudinal velocity variations, and their signals should attenuate significantly near the cliff due to the LOS geometry.

The large extra variation shown in Fig. 7 has a significant transverse component, i.e., along cross flow-line direction, thus it can not be mainly caused by surface melting, some other sources of forcing contribute more. We therefore studied points moving in a near-perpendicular direction to LOS, where along flow-line motion (e.g., velocity variation due to surface melting) is trivial in TRI data. The 2016 data is appropriate for this study.

We focused on three points in the mélange. The velocity estimates from both interferometry and feature tracking suggest

that their along-flow line velocities are almost perpendicular to the radar LOS direction (within $\pm 5°$ of $90°$). Any longitudinal variation would be trivial when projected onto the LOS direction. Figure 13 shows that the LOS velocity variation caused by up-and-down ice motion that is directly related to tides can only explain about half of the observed signal. The extra signal has a strong correlation with tidal rate, with an amplitude of $\sim 1$ m d$^{-1}$ ($\sim 0.1$ m in displacement). This phase relation suggests that either bed topography or tidal currents are responsible for the transverse signal. Bed topography is not likely to be the main

contributor, as it is more likely to affect glacier motion rather than mélange motion, unless mélange ice is strongly attached to the glacier. There is no ocean current record during our campaigns near the glacier front, and the available models are too coarse in the ice fjord. However, as Doake et al. (2002) have discussed, the usually accepted drag coefficient between ice and water is not likely to create enough force to drive ice motion to a sufficient magnitude. To fully explain the periodic transverse motion of ice, we need to either assume a very rough surface for ice below the water, so that ice motion driven by tidal ellipses





can be sufficient, or consider other sources of forcing. These forces are also likely to influence ice on the floating glacier tongue (Fig. 14). At a point on the glacier where ice moves ~90° to radar LOS, the TRI-derived velocity time series has a larger amplitude than the vertical tidal rate. This suggests that floating ice near the calving front in 2015 is weak, and moves in a similar behavior as the mélange ice.

## 5    Conclusions

High spatial and temporal resolution measurements of the time-varying velocity field at the terminus of Jakobshavn Isbræ were acquired with Terrestrial Radar Interferometry (TRI). Ocean tides modulate glacier velocity and this modulation can be used to infer the location of grounding line. The phase relation between ice velocity and tidal rate suggests a ~1 km wide floating zone in early summer of 2015 and 2016, where TRI-observed velocity variation contains ice up-and-down motion caused by

tide rise and fall, and perhaps transverse motion due to tidal currents. The floating zone moves together with calved ice through most of the calving season. However, in late summer 2012, there is no evidence of a floating ice tongue. We hypothesize that Jakobshavn Isbræ maintains a short floating tongue from winter to early summer, when ice flow exceeds ice loss by calving events, and the glacier front advances. In summer, iceberg calving surpasses ice flow, the glacier front retreats, and becomes nearly grounded by late summer. TRI-derived Digital Elevation Models support this hypothesis: in early summer, there is a

~1 km wide zone with relatively thin ice (<125 m) above local sea level; in late summer, ice thickness near the cliff increases dramatically and buoyancy is insufficient to support a floating glacier front.

*Competing interests.*    The authors declare that they have no conflict of interest.

*Acknowledgements.*    We acknowledge Denise Holland at the Center for Global Sea Level Change in New York University for organizing the field logistics in the campaign 2015 and 2016. Judy McIlrath of the University of South Florida is thanked for help in the 2012 fieldwork.

This research was partially supported by NASA grant NNX12AK29G to T.H. Dixon. D.M. Holland acknowledges support from NYU Abu Dhabi grant G1204, NSF award ARC-1304137, and NASA Oceans Melting Greenland NNX15AD55G. S. Xie thanks Nicholas Voss at the University of South Florida for the helpful discussions. Landsat-7/8 and Sentinel-2 images were downloaded through the USGS EarthExplorer. Envisat data were provided by the European Space Agency.





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



**Table 1.** Parameters used for TRI measurements

| Year | Start day | End day | Scanned arc (°) | Repeat time* (min) |
|------|-----------|---------|-----------------|--------------------|
| 2012 | 31 July | 12 August | 120 | 3 |
| 2015 | 6 June | 10 June | 150 | 1.5 |
| 2016 | 7 June | 20 June | 170 | 2 |

∗ Time between two adjacent scans.





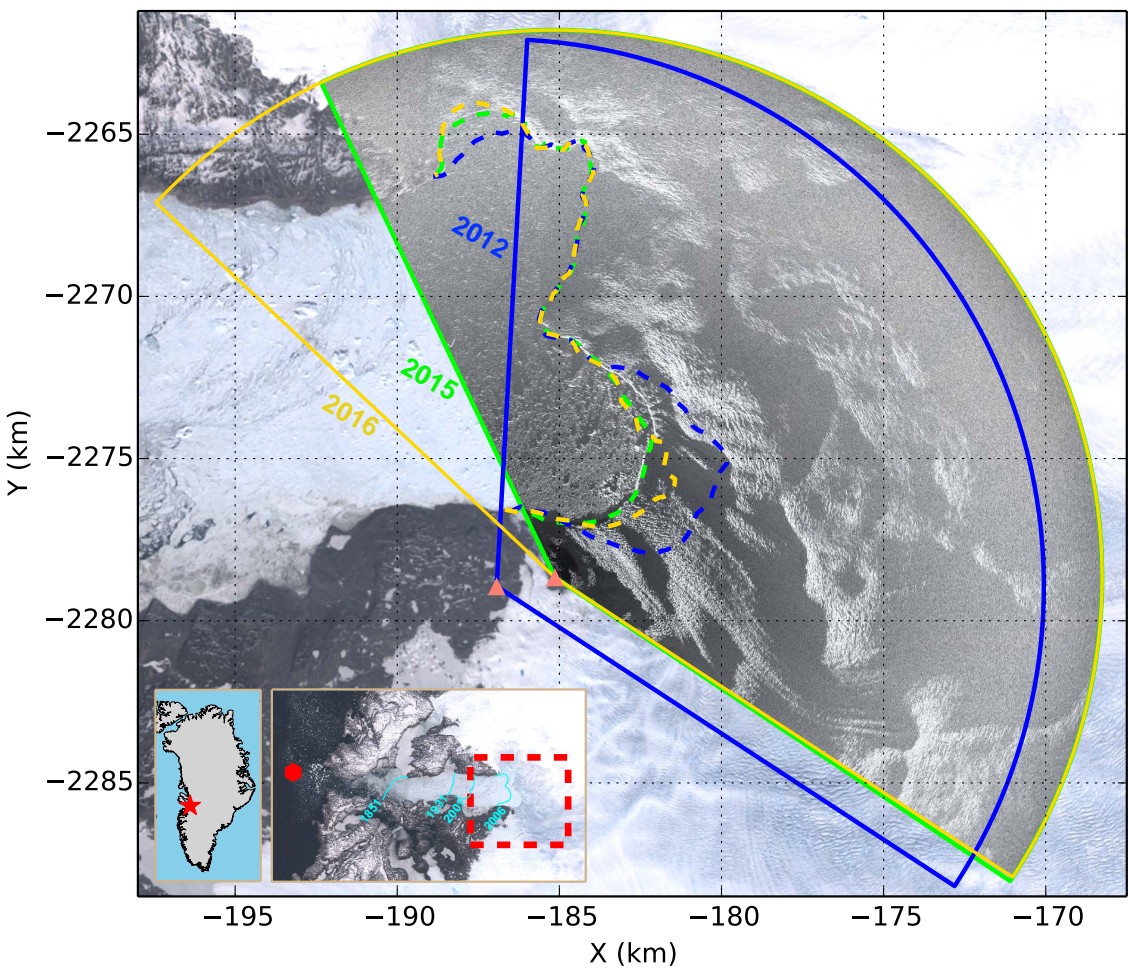

**Figure 1.** TRI scan areas in 2012 (blue), 2015 (green) and 2016 (yellow). An intensity image (acquired on 9 June 2015) is overlain on a Landsat-8 image (acquired on 4 June 2015). Dashed lines indicate the ice cliff locations from satellite images: Landsat-7 on 6 August for 2012; Landsat-8 on 4 June for 2015; and Landsat-8 on 13 June for 2016. Triangles in salmon color show locations of the radar. Dashed red box in the inserted figure outlines the area shown in the main figure. Cyan lines show the calving front positions in different years, redrawn courtesy of NASA Earth Observatory. Red hexagon marks the mooring location where tidal height was recorded in 2015. Red star shows the location of the study area in Greenland. Coordinates are in polar stereographic projection (EPSG: 3413).



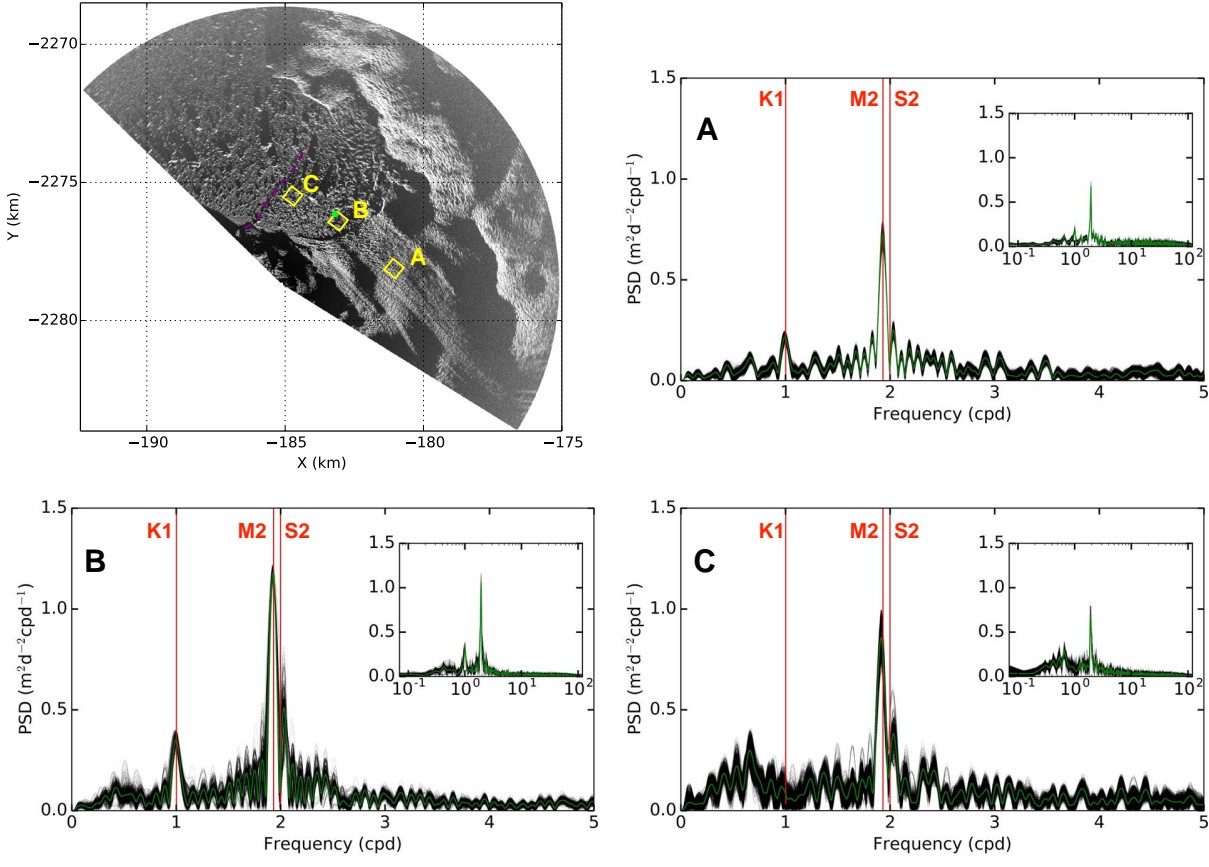

**Figure 2.** Stacked power spectral density (PSD) estimates of the LOS velocity time series for selected areas in 2016. Three 0.5 km × 0.5 km boxes (A, B, and C) mark the selected areas. For PSD plots, each black line represents 1 pixel (10 m × 10 m) in the corresponding box. Dark green line shows the mean value. PSDs shown here are normalized. Inserts in A, B and C show a wider range of spectra. Red lines mark frequencies of the K1, M2, and S2 tide constituents. On the upper left map, dashed purple line shows the observed height step-change in the mélange. Green dot shows location of the point used in Fig. 5.





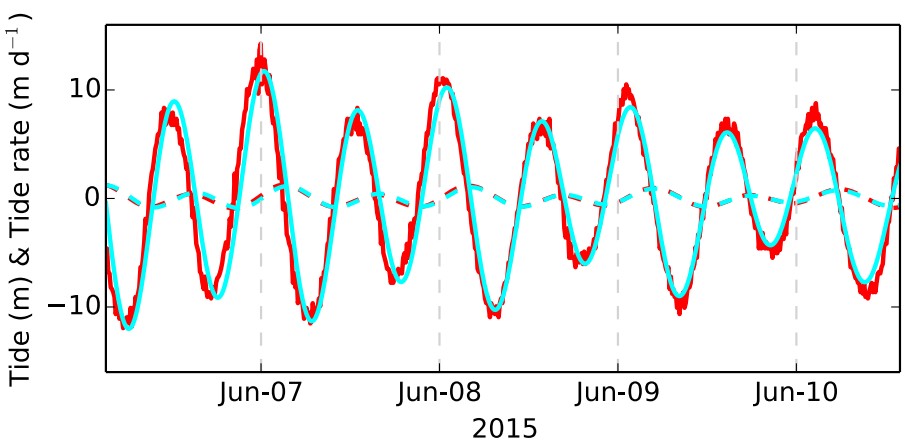

**Figure 3.** Predicted and observed tide and tidal rate during 2015 campaign. Tide is the dashed line, tidal rate is in solid line. Cyan color represents predicted tide and tidal rate, based on Richter et al. (2011). Red color represents observed tide and derived tidal rate from a mooring at the mouth of the fjord.



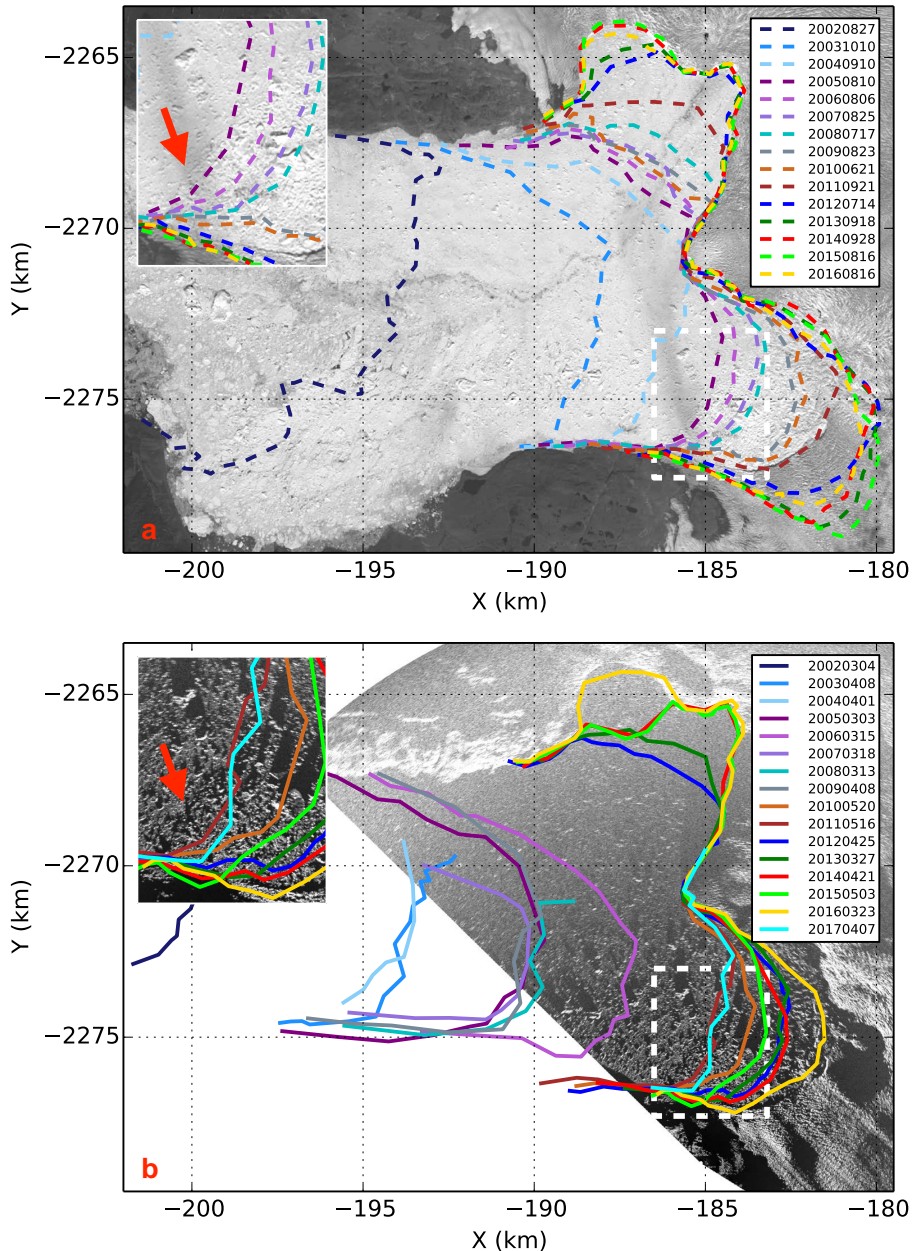

**Figure 4.** Step-change of mélange ice in 2016. Dashed white box outlines the inserted figure. Red arrows mark the step-change zone in the mélange, which can be seen from both Landsat-8 (a) and TRI (b) image acquired on 13 Jun 2016. Dashed and solid color lines show annual ice front minima and maxima with dates in the legend. Note that the ice cliff is sometimes indistinct when there is weak ice attached to the glacier that is nearly detached, we only show cliffs that are well defined by looking at satellite images. Cliff locations are derived from all available Landsat-7/8 and Sentinel-2 images except for 2005, when there were few clear images at Jakobshavn Isbræ due to cloudy weather, and Envisat images were used instead.




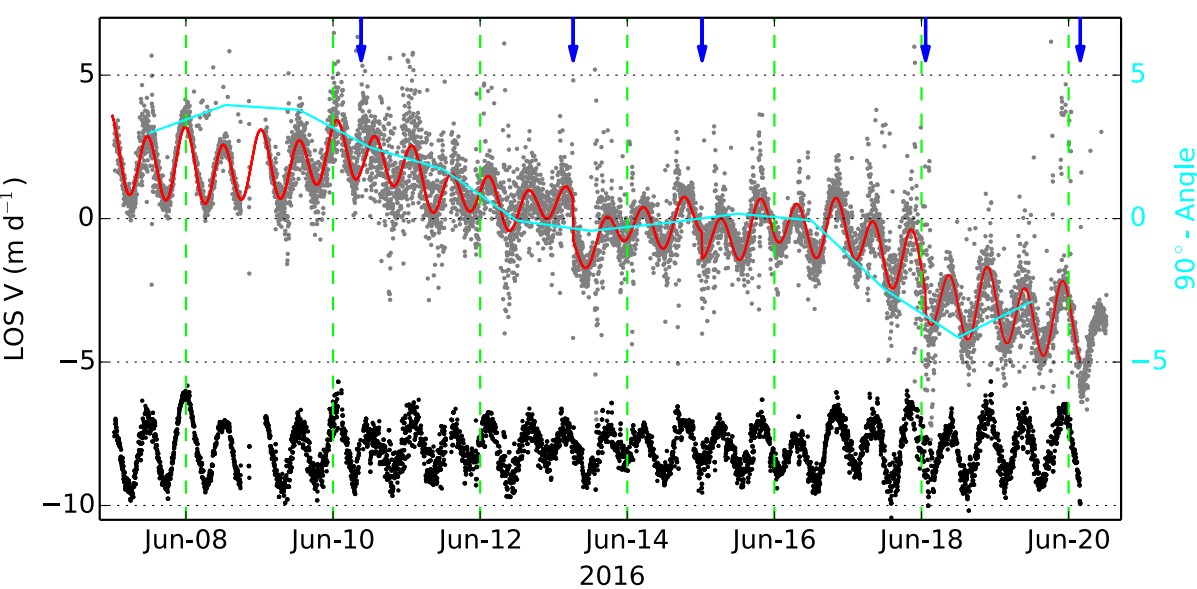

**Figure 5.** TRI observed LOS velocity time series for a single point in 2016. Grey dots show velocities derived from unwrapped phases, red curve shows the model used to remove perturbations caused by calving events, black dots show detrended time series offset by -8 m d$^{-1}$. Blue arrows mark major calving events. Cyan line shows changes of angle between LOS and 2-D ice velocity direction by feature tracking. The LOS velocity variation of period longer than 1 d is mostly due to changes of background velocity direction. Green dot on the map in Fig. 2 shows location of the point used for this figure.





**Figure 6.** Phase lag map and time series of a profile for each campaign. Grey dots (a, c, e) show detrended LOS velocity time series for a profile along the ice flow-line, marked by white dots on the right map. Red curve shows best model fit. LOS velocities are offset to be distinct from each other. Cyan curve shows tidal rate. Phase lag map (b, d, f) for M2 frequency signal. Areas where SNR<1.5 are omitted. Phase lags are converted to times (in h). Solid and dashed pink lines show maximum and minimum extents of glacier front for the corresponding year. In (f), dashed red line shows TRI derived location of ice cliff on 13 June 2016.



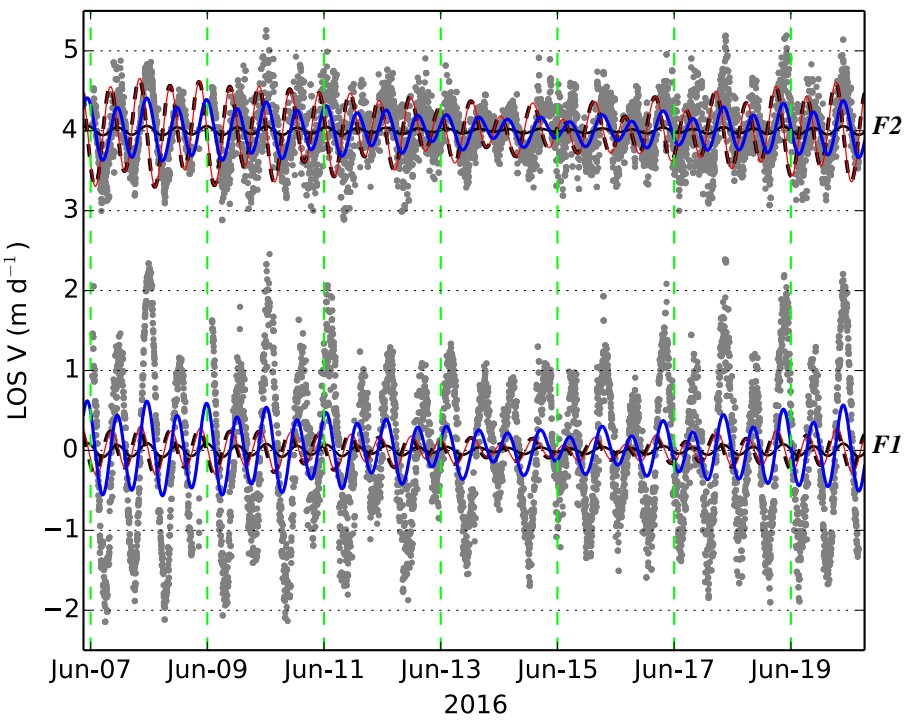

**Figure 7.** Detrended LOS velocities of points located in the low phase lag zone in 2016. *F1* and *F2* are the two points marked with purple triangles in Fig. 6(f), *F2* (upstream one) has been offset by 4 m d$^{-1}$ for illustration purposes. Grey dots are observed time series. Solid black curve shows vertical response to tide variations, using admittance of 0.15 (Podrasky et al., 2014) and is projected onto the LOS direction. Dashed black curve is horizontal response by using admittance of 0.12, projected onto the LOS direction. Red curve shows the sum of solid and dashed black curves, its Pearson correlation coefficient with observed time series is -0.13 and -0.19 for *F1* and *F2*, respectively. Blue curve shows predicted LOS velocity by assuming ice is free floating; its Pearson correlation coefficient with observed time series is 0.82 and 0.69 for *F1* and *F2*, respectively.





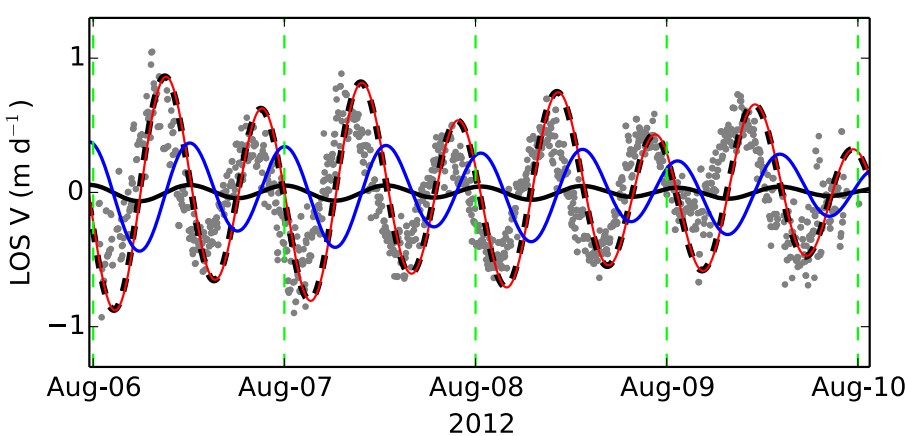

**Figure 8.** LOS velocities of a point immediately adjacent to the ice cliff in 2012 (*B1* in Fig. 6(b)). Colors and curves represent the same parameters as in Fig. 7. The Pearson correlation coefficient with observed time series is 0.65 for the red curve (grounded or nearly grounded assumption, the same as Podrasky et al. (2014)), and -0.56 for the blue curve (free floating assumption).





**Figure 9.** DEM for the glacier front, derived from a 1 day median average. For each subplot, pink contours show bed bathymetry (An et al., 2017). Dashed red line shows the ice cliff from TRI image, note that in 2016 it was not possible to distinguish a portion of the ice cliff from TRI measurements, hence it is not marked on the map. The background image for (a) was acquired on 6 Aug 2012 by Landsat-7, white stripes are data gaps. Background image in (b) was acquired on 4 Jun 2015 by Landsat-8. Background image in (c) was acquired on 13 Jun 2016 by Landsat-8. Note that uncertainty increases with distance to the radar. Blue, green, and yellow line in (d) shows an elevation profile along a transect marked by grey line in (a–c), respectively. These transects have the same location in space. In (e), the distance of each transect is normalized so that the cliffs are in the same position.





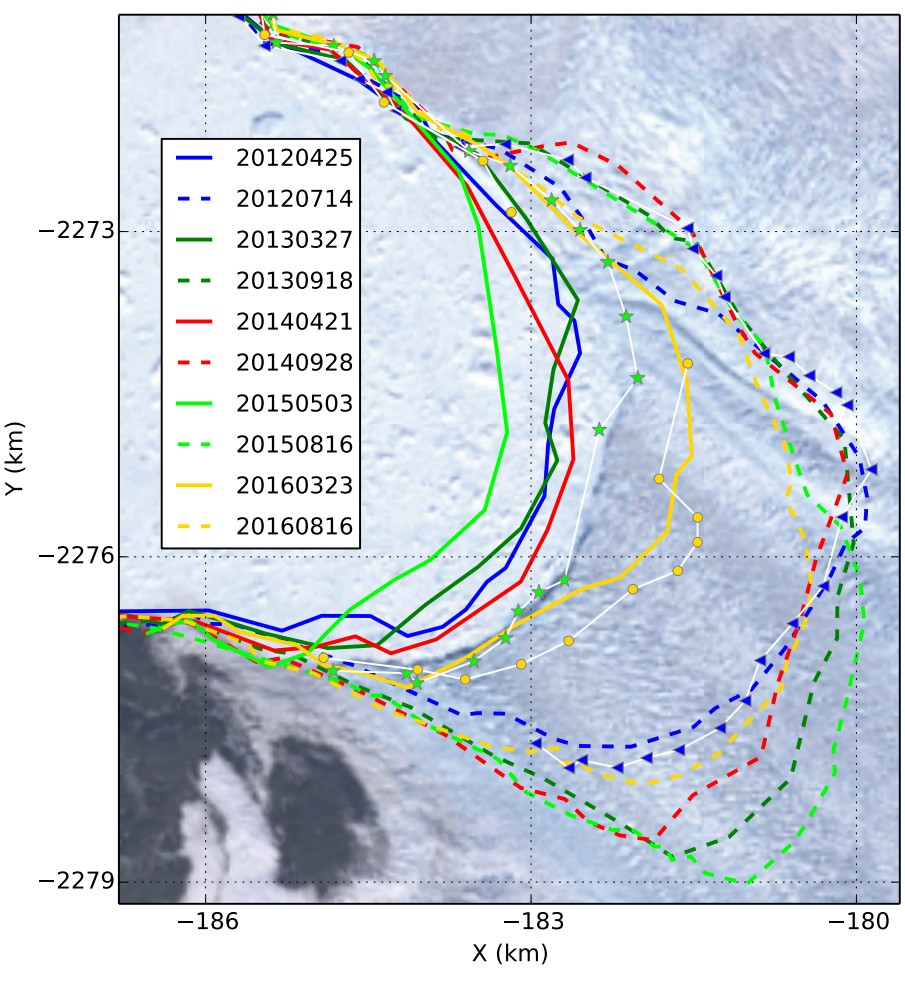

**Figure 10.** Annual maximum and minimum extents of Jakobshavn Isbræ's calving front from 2012 to 2016. Solid lines show the ice cliff when glacier extent is maximum, dashed lines when glacier extent is minimum. Ice cliff locations are derived from all available Landsat-7/8 and Sentinel-2 images from USGS archive. Legends are dates of images acquired by satellites. Lines with triangles, stars and circles show ice cliff locations during TRI campaigns in 2012 (6 Aug), 2015 (9 Jun) and 2016 (13 Jun), respectively. Background for this figure is a Landsat-8 image acquired on 4 June 2015.



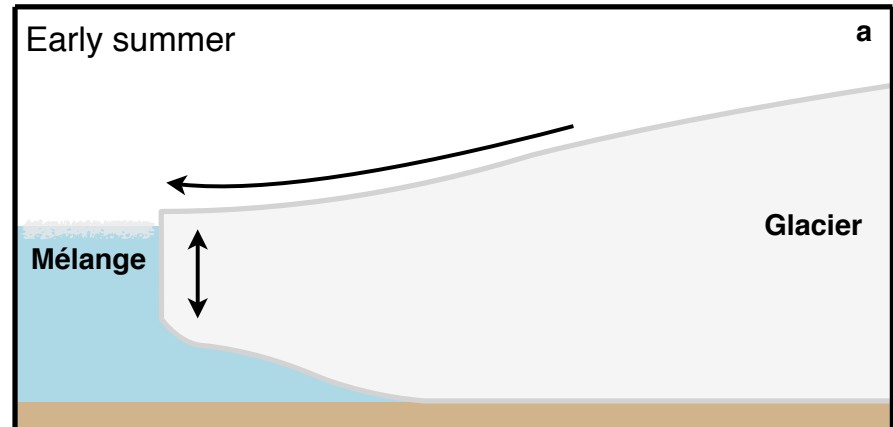

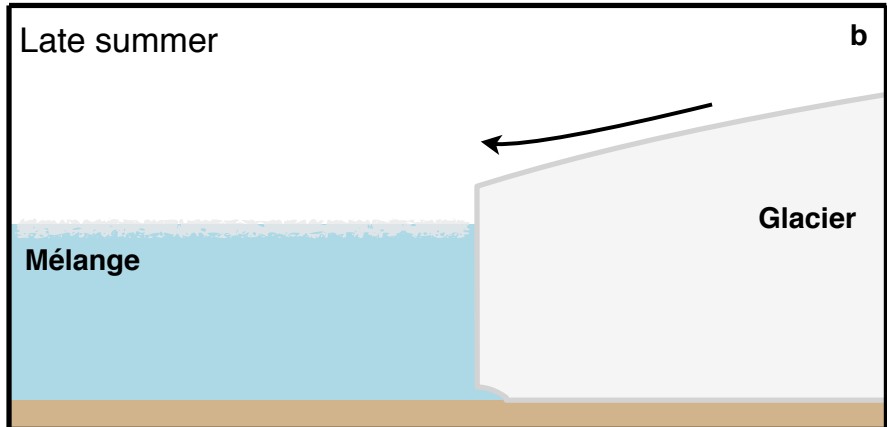

**Figure 11.** Cartoon of glacier front in early (a) and late (b) summer. Ice shown in light grey, water shown in light blue, bed shown in brown. Single-sided arrow indicates direction of glacier flow. Double sided arrow in (a) indicates that ice tongue moves in phase with tides. Note that the ice cliff is higher in late summer than early summer.





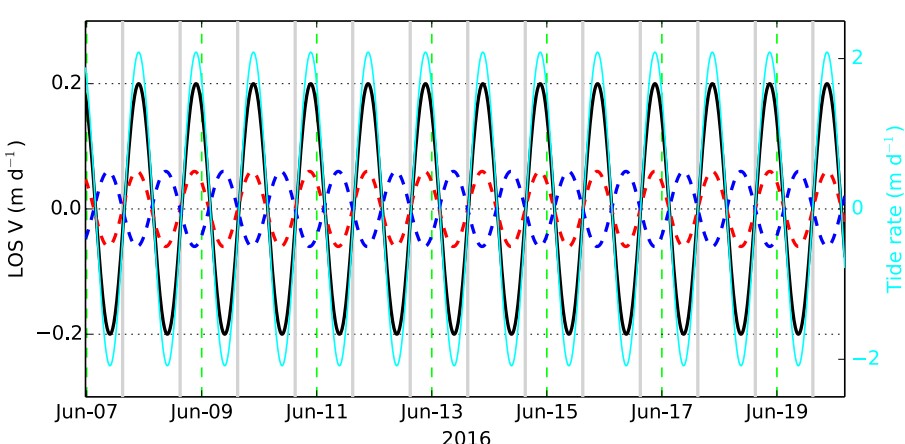

**Figure 12.** Possible diurnal velocity variation caused by surface melting in 2016. Cyan curve shows ocean tidal rate of the K1 ocean tide. Black curve is the synthetic diurnal signal correlated with the tidal rate (phase lag = 0). Grey lines mark local noon. Dashed curves show velocity variation due to surface melting (not to scale, assuming speed maxima lag local noon by 6 h), projected onto LOS direction. Red lines show motion towards the radar and blue lines show motion away from the radar.





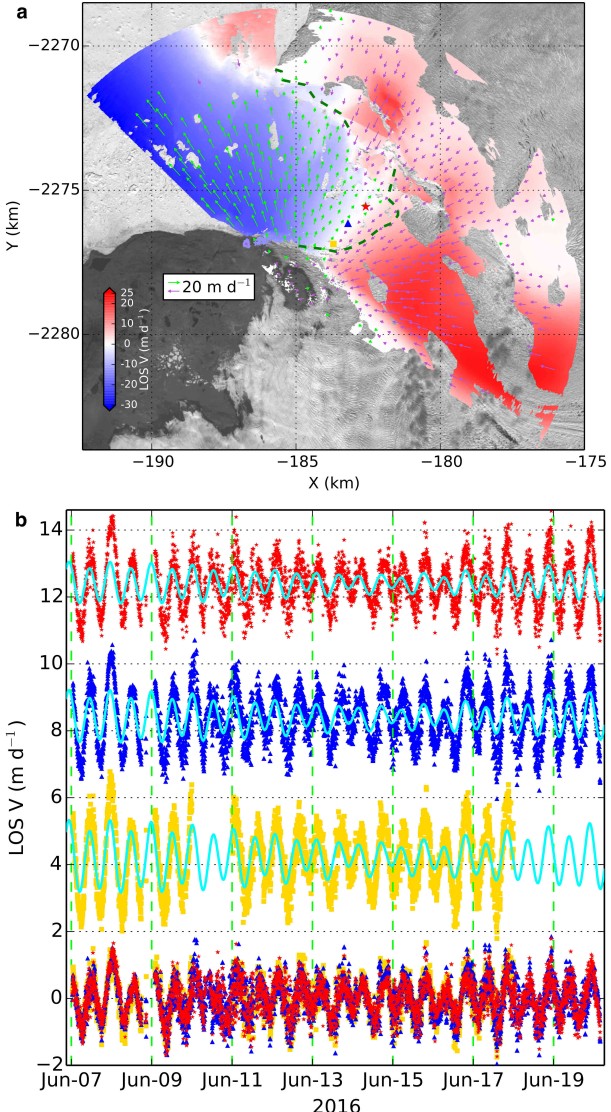

**Figure 13.** Transverse ice motion in the mélange of 2016. (a), Color map shows LOS velocity by interferometry, from a 1 day median average. Arrows show velocity estimates from feature tracking projected onto the LOS direction (purple when ice moves toward the radar and green when moves away). Dashed dark green line shows cliff location from TRI image. Yellow square, blue triangle and red star mark three points, where their 2-D velocity direction is nearly perpendicular to radar LOS. Their LOS velocity time series are shown in (b). Top 3 rows (b) show TRI observed LOS velocities for selected points, corresponding to the same marker as on map (a); cyan curves are predicted LOS velocities based on the geometry, assuming ice is free floating. LOS velocities are offset by some values for visual purposes. The bottom row shows residual time series by subtracting the cyan curves.





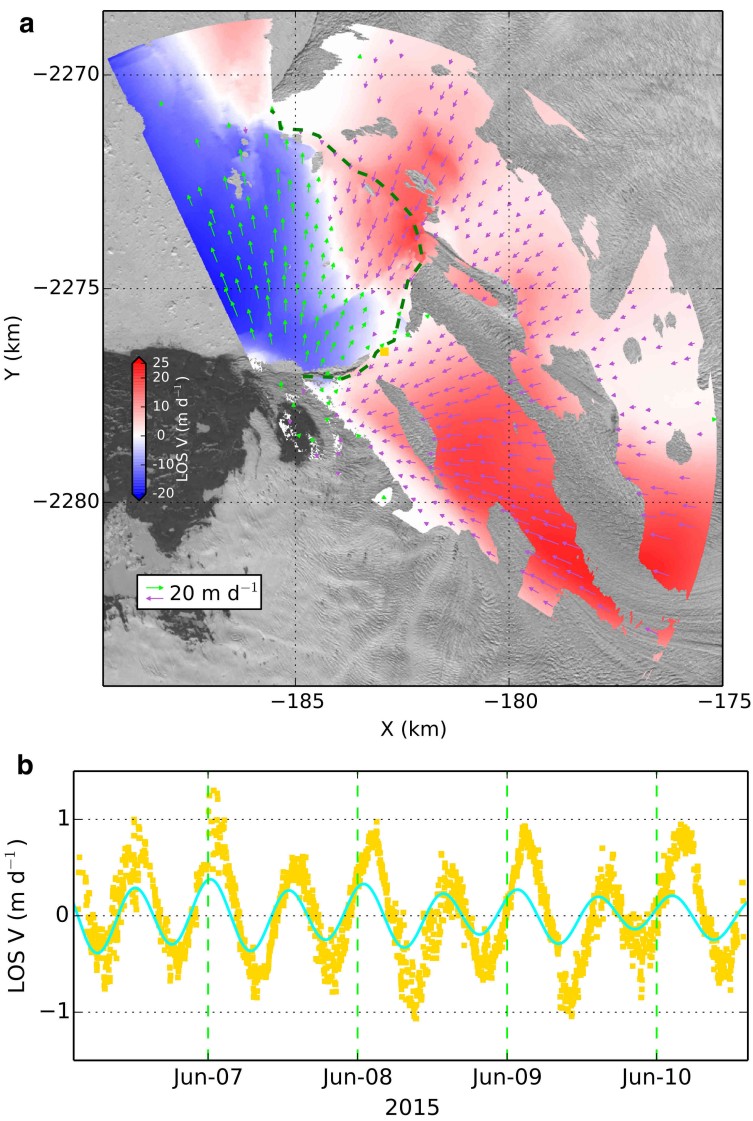

**Figure 14.** Transverse ice motion on the glacier front for 2015. Colors and arrows represent the same parameters as in Fig. 13. A point immediately adjacent to the cliff was chosen, marked by yellow square on (a) with its LOS velocity observed with TRI and predicted by tide variations are shown in (b). Cyan curve (b) shows predicted LOS velocities, yellow dots show observed time series at the point marked by yellow square in (a).