# Peer review of "Grounding line migration through the calving season at Jakobshavn Isbræ, Greenland, observed with terrestrial radar interferometry"

_The Cryosphere, 2017_

## Referee Comment (RC1) · Anonymous Referee #1 · 2 Feb 2018

The paper presents terrestrial radar interferometry (TRI) measurements from Jakobshavn calving front. Three season of field measurements (measuring from 4 days to almost 2 weeks) of velocities and digital terrain models are presented. These radar data give new documentation/verification the dynamics of the calving front. Physical challenges and dangers connected to field measurements in the calving area are well known, and this project is a valuable contribution to possible future development of measuring programs for increased knowledge of calving dynamics. The dynamic of the mélange in front of Jackobshavn calving front is one aspect that can make measurements demanding. The paper describes and discuss the calving cycle, with advance of the glacier front which forms a floating ice tongues during winter, and the retreat

of the tongue by calving during summer. The data set documents the grounding line migration during the calving season from velocities clearly modulated by tides (well presented in fig. 6), and thus the flotation of the calving front in a convincing way.

The data analysis is thoroughly, and the paper is well written. The paper clearly demonstrates the potential of radar monitoring of calving events, which is relevant due to expected increased in calving activity due to global warming with warmed oceans.

The paper is very well written, with clear language, relevant references, good method description and uncertainty discussions. It provides an interesting discussions of the dynamics of the melange on p. 6 , l.4 .

The only concern are the relevance of the very many figures, both in paper and supplementary text. It seems that the main figures are Fig 1, 2, 6, 9 and 13.

I suggest the authors consider removing all the other figures, and possibly try to simplify the figures they keep, and maybe combine differently and simplify the information here. The paper must then be slightly rewritten – where referring to the figures.

Specific comments: On p. 4, l. 1, Other errors in TRI data, such as phase variations associated with variable atmospheric water vapor, are difficult to model. Is this true? Corrections of refraction could be calculated from meteorological data if available?

Fig. 1, caption line 1, An intensity image.. (Specify: intensity of radar backscatter from your own measurements?)

Fig. 2. Inserts in A,B,C, necessary info?

Fig. 3 – move to supplementary material?

Fig. 9 DEM from glacier front, derived from a one day average (please specify average of what)

Fig. 11. Necessary for readers of the Cryosphere? Quite simple principle.

Fig. 13 and 14, combine to one figure?

---

## Referee Comment (RC2) · Anonymous Referee #2 · 15 Feb 2018

This study analyzed terrestrial radar interferometry data collected at Jakobshavn Isbrae during field campaigns in 2012, 2015, and 2016. Through tidal analysis of line-of-sight velocities, the authors conclude that the terminus is floating in early summer and becomes increasingly grounded as the terminus retreats throughout the summer. These observations build on previous work from Jakobshavn Isbrae and elsewhere, which together provide a consistent picture of terminus morphology. These observations provide important insights into the processes influencing iceberg calving rates.

The observations in the paper are good, but the paper could use some editing to improve clarity and focus. For example, although interesting, the discussion of ice

melange is not really relevant to the paper as written. I suggest either removing the discussion of ice melange or better integrating it into the text. The paper is about grounding line migration, but then there are some sentences and paragraphs about ice melange that are sprinkled throughout the manuscript but that don't really fit with the rest of the paper.

The paper also isn't particularly long, which makes me wonder why additional text is needed in supplementary material. Couldn't it be incorporated into the manuscript? The portion about the feature tracking seems to be important, but is only briefly mentioned in the main text.

Some of the figures could also use work: 1. The green lines in Figure 2 are almost undetectable, and the red and green lines will be difficult for some readers.

2. Figure 6 is also really difficult to read, and it may be misleading in that the tidal response appears to grow in both the upglacier and downglacier directions, but is minimal somewhere in between. I understand that this is at least partially due to flow direction not corresponding with line of sight, but that needs to be made more clear. You could indicate that the smallest response occurs where the flow is perpendicular to line of sight.

3. Figure 9: It would be nice to see the location of the radar on these maps. Its pretty obvious where its located in the MLI images in previous figures, but not in the DEMs.

4. Figure 12: I'm not sure what purpose this figure really serves.

5. Figure 13: Its really difficult to see the arrows in panel a. And why are feature tracking velocities projected onto line of sight? That seems backward and misleading to me, as it gives the impression that velocity variations are toward/away from the radar. I think it would be better to project the LOS velocities into map view, and then talk about what causes the variations in "true" velocity.

Some specific comments: Page 1, Line 5: I would not say that ice is locally thin if the

freeboard is less than 125 m!

Page 1, Line 22: "down dipping upstream bed" is confusing. Do you mean retrograde bed?

Page 2, Line 2: "ice speed accelerates" – speed doesn't accelerate, but ice does

Page 2, Line 15: "grounding line position" is not really a "basal condition"

Page 2, Line 30: "through a calving season"? This makes it sound like you were operating the TRI all summer long, which is misleading.

Page 5, Line 19: "The amplitude" of? I'm not quite sure what this refers to.

Pages 5-6 (and supplement): The step-change in ice melange thickness is interesting and suggests that the ice melange has a "terminus". However the discussion is highly speculative and doesn't really fit in this subsection, which is about tidal analysis. Also, I don't buy the idea that the change occurs because of some bedrock topographic feature. I wonder if instead you are seeing the remnants of the winter melange that hasn't yet lost cohesiveness.

Page 7, Lines 19-20: This seems like a pretty big assumption, considering that other studies have suggested year-to-year variability in tidal response.

Page 8, equation 3: Double check this equation. I'm pretty sure that V_los and dh/dt should be swapped.

Page 8, Line 17: dh/dt $\sim$ 0.1*tidal rate in the melange due to buoyancy effects, and less than that for the glacier.

---

## Author Comment (AC1) · 26 Feb 2018

**Responses to anonymous referee #1:**

We appreciate comments from this reviewer on our manuscript. Responses we made are below in red.

The paper presents terrestrial radar interferometry (TRI) measurements from Jakobshavn calving front. Three season of field measurements (measuring from 4 days to almost 2 weeks) of velocities and digital terrain models are presented. These radar data give new documentation/ verification the dynamics of the calving front. Physical challenges and dangers connected to field measurements in the calving area are well known, and this project is a valuable contribution to possible future development of measuring programs for increased knowledge of calving dynamics. The dynamic of the mélange in front of Jackobshavn calving front is one aspect that can make measurements demanding. The paper describes and discuss the calving cycle, with advance of the glacier front which forms a floating ice tongues during winter, and the retreat of the tongue by calving during summer.

The data set documents the grounding line migration during the calving season from velocities clearly modulated by tides (well presented in fig. 6), and thus the flotation of the calving front in a convincing way.

The paper is very well written, with clear language, relevant references, good method description and uncertainty discussions. It provides an interesting discussions of the dynamics of the melange on p. 6 , l.4 .

The data analysis is thoroughly, and the paper is well written. The paper clearly demon- strates the potential of radar monitoring of calving events, which is relevant due to expected increased in calving activity due to global warming with warmed oceans.

The only concern are the relevance of the very many figures, both in paper and supplementary text. It seems that the main figures are Fig 1, 2, 6, 9 and 13.

I suggest the authors consider removing all the other figures, and possibly try to simplify the figures they keep, and maybe combine differently and simplify the information here. The paper must then be slightly rewritten – where referring to the figures.

We have removed and combined some figures, the revised manuscript have 9 figures, and we have rewritten places where referring to the revised figures.

Specific comments: On p. 4, l. 1, Other errors in TRI data, such as phase variations associated with variable atmospheric water vapor, are difficult to model. Is this true? Corrections of refraction could be calculated from meteorological data if available?

In theory, if enough meteorological data were available, some corrections could be applied. However, in our case, it is difficult to do such corrections. There are two main reasons: 1) No well distributed stationary points are available to define a model, because we only have very limited near-field areas with targets that are not moving, and rocks points on the other side of the fjord have much lower coherence than the interested area. Points on ice can not be used to define a correction model because they are treated as kinematic targets. 2) Water vapor content varies significantly on space. During our campaigns, clouds or dense fog were sometimes seen in front of the ice cliff, but not other places. Besides, such errors should not change measured velocities significantly in the near-field, because the interferograms were formed between adjacent scans separated by 1.5-3 min. Phase variations due to changes of water vapor should be small in this short time span compared to the relatively large variations caused by fast ice motion. We therefore did not correct errors related with atmosphere. Instead, we analyzed data <10 km of the radar to minimize water vapor effects. In addition, we omitted data with SNR<1.5 in our tidal response analysis.

We have rewritten this sentence to state "Other errors in TRI data, such as phase variations associated with variable atmospheric water vapor between adjacent scans, are difficult to model but should not be significant in the nearfield given the 1.5–3 minute repeat time".

Fig. 1, caption line 1, An intensity image.. (Specify: intensity of radar backscatter from your own measurements?)

Yes, this is an intensity image of radar backscatter from our measurement. We have added this information to the caption of Fig. 1 as "An intensity image of radar backscatter from the 2015 campaign (acquired 9 June 2015) is overlain on a Landsat-8 image (acquired 4 June 2015)".

Fig. 2. Inserts in A,B,C, necessary info?

They were used to show a wider range of frequency, we have removed these inserts.

Fig. 3 – move to supplementary material?

Done.

Fig. 9 DEM from glacier front, derived from a one day average (please specify average of what)

This figure has now become Fig. 7. We have modified this sentence to "DEM for the glacier front, derived from median average of DEM estimates separated by 2 minutes during a 1 day period".

Fig. 11. Necessary for readers of the Cryosphere? Quite simple principle.

This figure has now become Fig. 8. We think some readers may still be interested in it.

Fig. 13 and 14, combine to one figure?

Done. See Fig. 9.

[revised manuscript text omitted]

**Contents:**

Figure and movie contents are introduced in the text.

**1. Method to fix phase offsets in the 2012 data**

Ice velocity estimates have some significant offsets in the 2012 data. These offsets represent phase discontinuities, clustering at integer multiples of the radar wavelength (Fig S1(a)). They occur because the 2012 data were acquired with a scan rate that was slow compared to ice velocity, such that ice could move more than one radar wavelength relative to adjacent area during the 3 minute scan interval (data in later years were acquired with a faster scan rate). Compared to fast moving ice, rocks near the TRI can be treated as stationary objects. We used a rock reference point ~0.2 km away from the radar as a reference for phase unwrapping. To correct the phase discontinuities caused by phase unwrapping error, we firstly used a model to fix the phase offsets for a single point on ice (P0, yellow point in Fig S1(b)), then used it as reference point for phase unwrapping to get velocity maps relative to this point. Ice velocities relative to the stationary rock were then generated by adding the modeled velocities of P0 to those relative velocities. We selected data obtained between 6 August and 10 August for the following process because there were continuous measurements during this period and only one small calving event. Here are the 4 steps used to model the velocity of P0:

1) Use the same data processing procedure as described in *Voytenko et al.* [2015)] and *Xie et al.* [2016], generating time series for all velocity maps. The generated velocities are absolute velocities, plus/minus some value equal to an integer number of wavelength jumps. The TRI instrument transmits Ku-band microwaves with wavelength of 1.74 cm. In a 2-way measurement system, 1 wavelength jump in unwrapped phase leads to 0.87 cm change in the LOS displacement, equal to 4.2 m d$^{-1}$ offset in LOS velocity when the repeating time of measurement is 3 minutes. From Fig S1(a) we infer that most of the unwrapped phases for P0 have 4 cycles of phase offset.

2) Remove "apparent outliers" by using the modified Z-score method [*Iglewicz and Hoaglin,* 1993]. For the *i*th observation $x_i$, its Z-score is:
$$Z_i = 0.6745(x_i - \tilde{x}) / \text{MAD} \qquad (S1)$$
where MAD denotes the median absolute deviation, and $\tilde{x}$ is the median value. Observations with absolute value of the modified Z-scores >3.5 were considered as outliers and removed. Then we subtracted a 2nd-order polynomial curve to remove the possible response to calving events. The method by *Davis et al.* [2014] was then used to estimate the periodic components caused by tidal variations. We chose 3 sinusoids with the same frequencies as the K1/M2/S2 tidal constituents (see the main paper for more information about the tidal constituents at Jakobshavn Isbræ). The solid blue curve in Fig S1(c) shows the best fit of a 2nd-order polynomial + 3 sinusoids to the observed time series. The dashed blue curves mark 3 times the value of the root-mean-square (RMS) of the residuals.

3) Shift all observations (including "apparent outliers" in step 2) upwards/downwards by 4.2×N m d$^{-1}$, where N = 0, ±1, ±2, ±3, ±4, ±5, etc. Observations who's shifted values do not fall into the 3×RMS space defined in step 2 are labeled as outliers (grey dots in Fig S1(c)) and removed. Values that fall within the 3×RMS space are then shifted by 4 cycles of phase offset (16.8 m d$^{-1}$) to eliminate the jumps derived by feature tracking method in

step 1.

4) Apply a median filter (kernel size = 3) to the time series from step 3, and then use the same model in step 2 to fit the new time series. Estimated parameters are then used for further processing. Fig S1(d) shows the least square fit of the final fixed time series for P0.

For all other points on the TRI image, their velocities are estimated by adding the modeled velocity of P0 to their relative velocities. Fig S2 shows phase offsets-fixed LOS velocities for selected points. Black dots are velocity estimates when using a stationary rock point as the reference point for phase unwrapping (shifted upwards by 4 cycles of phase offsets). From Fig S2 we see that phase jumps are greatly reduced by this method, especially on the glacier. In the mélange, time series shortly after two calving events (on 5 August and 9 August, see blue arrows in Fig S1(a)) still have some jumps, mainly due to phase breaks caused by rapid ice motion after calving events. In the tidal analysis section of the main paper, we omit data for the mélange acquired near these 2 calving events.

We used feature tracking (done with OpenCV: http://opencv.org/) as an independent method to examine the phase offset "fixed" velocities from interferometry. And used stationary or near-stationary points near the radar as references to calculate uncertainties in feature tracking, which is <1 m d$^{-1}$ in both x and y components for velocity estimates based on a TRI intensity image pair separated by 1 day. Fig S3(a) is the median LOS velocity map from a 1 day sequential measurements in the 2012 campaign. Fig S3(b) shows a velocity map (projected onto the LOS direction) derived by feature tracking using two TRI intensity images acquired at the beginning and end of the day (red means moving towards the radar, yellow means moving away). Fig S3(c) is the difference between (a) and (b). Red means the velocity estimated by interferometry is larger than the velocity estimated by feature tracking, yellow means it is less. For most points, the difference is much smaller than 1 cycle of phase jump (4.2 m d$^{-1}$). On the north edge of the mélange, there is a small area where offsets still exist, presumably caused by phase breaks due to discontinuities in the TRI maps. One way to solve this type of phase jump is to use shorter repeat time when collecting data. However, this problem won't affect the following tidal analysis if the sign of LOS velocity has not been changed by these phase jumps. Because this study focuses on short time-scale tidal responses, the large background velocity was detrended before tidal analysis. We also noticed that the velocity differences near the radar are much larger, up to a few cycles of phase jumps. This is due to the fact that the phase data for stationary areas near the radar have no discontinuity problems in speed as on the ice. The method described above has thus introduced some artificial jumps in the stationary near field.

Fig S4 and S5 show comparisons of velocity estimates by interferometry and feature tracking for 2015 and 2016. There are no systematic differences for these two years. Except for some random errors, significant differences only appear in isolated patches where the phase map lacks continuity, or in places close to where calving events have occurred. Thus, we did not apply the same method used for 2012 to the 2015 and 2016 data. Instead, we did phase unwrapping by using stationary points on rocks only.

**2. PSD plots for 2012 and 2015**

Fig S6, power spectral density (PSD) for selected areas in 2012
Fig S7, power spectral density (PSD) for selected areas in 2015

**3. Tide and tidal rate**

Fig S8 shows predicted and observed tide and tidal rate during 2015 campaign. In this study, tidal rate is defined as the 1st time derivative of tidal height. For a tide signal:
$$H(t) = A \cos(2\pi f t + \varphi) \tag{S2}$$
where $A$ is the amplitude, $f$ is the frequency, $\varphi$ is the phase. The tidal rate is:
$$H'(t) = 2\pi f A \cos(2\pi f t + \varphi + 0.5\pi) \tag{S3}$$
Compare to tidal height, the amplitude of tidal rate has been amplified by $2\pi f$. Due to differentiation, the phase difference between ice velocity and tidal rate is the same as the phase difference between ice position and tidal height.

**4. Phase lag maps for K1 and S2 tidal frequencies**

Fig S9, phase lag in time (h) for K1 and S2 tidal frequencies.

**5. Geometry between velocity and radar LOS**

Fig S10, geometry used to project velocity onto radar LOS direction.

**6. Movies shown major calving event during 3 campaigns**

Mov S1, Major calving events in 2012.
Mov S2, Major calving events in 2015.
Mov S3, Major calving (-like) events in 2016.

[Figure]

**Figure S1.** (**a**), LOS velocity time series (black dots) of P0 on (b), when using a stationary point on rock as reference for phase unwrapping. Pink bars show velocities by feature tracking method (shifted downwards by phase jumps of 4 cycles of radar wavelength), separated by ~1 d. Blue arrows mark visible calving events on TRI intensity images, two light blue arrows represent calving events that were in observation gaps. (**b**), Point (P0) chosen as reference point for phase unwrapping to estimate relative velocities. Red arrows show 2-D velocity map by tracking two TRI intensity images between 00:01:00 7 August and 00:00:00 8 August. Background is a TRI intensity image acquired on 6 August 2012. Arrows within the dashed white arc are used to assess the uncertainty of feature tracking, the RMS is <1 m d$^{-1}$. (**c**), LOS velocity time series between 6 August and 10 August (dots between two grey lines in (a)). Solid blue curve shows the best fit after "apparent outliers" (defined by modified Z-score method) removed, by using a 2nd-order polynomial + 3 pairs of sinusoids model. Dashed blue curves show 3 times RMS space of the residuals. Solid and dashed Light blue curves are blue curves moved upwards/downwards by integer numbers of radar wavelength cycle. Dots fall into the 3×RMS spaces (red) were used in (d), grey dots were removed as outliers. (**d**), Time series from (c) after shifting upwards by phase jump of 4 cycles of radar wavelength, and applying a median filter with a window-size of 3 (equals to 9 min in this case, where the repeating time of measurement is 3 min). Upper blue curve shows the best fitting by the same model as (c), but used filtered time series. Lower blue curves show the 2nd-order polynomial and sinusoids components, and are offset for clarity. Cyan curve shows local tidal height rate prediction.

[Figure]

**Figure S2.** LOS velocities for selected points, after phase jumps fixed. (**a**), Point locations for P1 - P8. (**P1-P8**), Red dots are phase jumps fixed time series, black dots are time series when choosing a stationary point as reference for phase unwrapping, and were shifted upwards by 4 cycles of phase jumps.

[Figure]

**Figure S3.** Comparison between velocity estimated by interferometry and feature tracking. (**a**), Median LOS velocity map by interferometry method for 7 August 2012, after phase jumps fixed. Background is a Landsat-7 image acquired on 6 August 2012, white stripes are data gaps. (**b**), Velocity map by feature tracking method, projected onto radar LOS direction. Red moves towards the radar, yellow away. (**c**), Difference between (a) and (b), red when velocity by interferometry is larger than by feature tracking, yellow vice versa. 4.2 m d⁻¹ equals to 1 cycle of phase jump.

[Figure]

**Figure S4.** Comparison between velocity estimated by interferometry and feature tracking in 2015. (**a**), LOS velocity map by interferometry. (**b**), 2-D velocity map by feature tracking. Arrows within the dashed white arc are used to assess the uncertainty of feature tracking, the RMS is <1 m d$^{-1}$. (**c**), Velocity map by feature tracking method, projected onto radar LOS direction. Red moves towards the radar, yellow away. (**d**), Difference between (a) and (c), red when velocity by interferometry is larger than by feature tracking, yellow vice versa. 8.4 m d$^{-1}$ equals to 1 cycle of phase jump, when repeating time was 1.5 min.

[Figure]

**Figure S5.** Comparison between velocity estimated by interferometry and feature tracking in 2016. Color and arrow denote the same as Fig. S3. In (c), 6.3 m d$^{-1}$ equals to 1 cycle of phase jump, when repeating time was 2 min. The 2-D velocity map is shown in Fig. 3(a) of the main paper.

[Figure]

**Figure S6.** Stacked power spectral density (PSD) estimates of the LOS velocity time series for selected areas in 2012. Two 0.5 km × 0.5 km boxes (A and B) mark the selected areas. For PSD plots, each black line represents 1 pixel (10 m × 10 m) in the corresponding box. Yellow lines show the mean value. PSDs shown here are normalized. PSD analysis was done by using the Lomb-Scargle method (Lomb, 1976; Scargle, 1982). Red lines mark the frequencies of K1, M2, and S2 tide constituents.

[Figure]

**Figure S7.** PSD plot for selected samples in 2015. Lines and colors denote the same as in Fig. S6.

[Figure]

**Figure S8.** Predicted and observed tide and tidal rate during 2015 campaign. Tide is the dashed line, tidal rate is in solid line. Cyan color represents predicted tide and tidal rate, based on Richter et al. [2011]. Red color represents observed tide and derived tidal rate from a mooring at the mouth of the fjord.

[Figure]

**Figure S9.** Phase lag maps of signals with K1 and S2 frequencies for each campaign. (a-b) for 2012, (c-d) for 2015, (e-f) for 2016. (a, c, e) show phase lags in time (h) for K1 frequency signal. (b, d, f) for S2 frequency signal. areas where SNR<1.5 are omitted.

[Figure]

**Figure S10.** Geometry used to project velocity onto radar LOS direction. Blue dot represents the radar, red square shows the mean vertical position of the target, red dot is the vertical position at current time. $H_0$ is the mean height difference between the radar and the target. $L$ is the horizontal distance between the radar and the target. $h$ is the vertical movement relative to $H_0$. $dh/dt$ is the vertical component of ice velocity. $V_{los}$ represents the TRI-observed LOS velocity. $V_{perp}$ represents ice velocity projected onto perpendicular direction of the radar LOS.

---

## Author Comment (AC2) · 26 Feb 2018

**Responses to anonymous referee #2:**

We appreciate comments from this reviewer on our manuscript. Responses we made are below in red.

This study analyzed terrestrial radar interferometry data collected at Jakobshavn Isbrae during field campaigns in 2012, 2015, and 2016. Through tidal analysis of line-of-sight velocities, the authors conclude that the terminus is floating in early summer and becomes increasingly grounded as the terminus retreats throughout the summer. These observations build on previous work from Jakobshavn Isbrae and elsewhere, which together provide a consistent picture of terminus morphology. These observations pro- vide important insights into the processes influencing iceberg calving rates.

The observations in the paper are good, but the paper could use some editing to improve clarity and focus. For example, although interesting, the discussion of ice melange is not really relevant to the paper as written. I suggest either removing the discussion of ice melange or better integrating it into the text. The paper is about grounding line migration, but then there are some sentences and paragraphs about ice melange that are sprinkled throughout the manuscript but that don't really fit with the rest of the paper.

We have removed the discussion of ice melange that is beyond the scope of this paper.

The paper also isn't particularly long, which makes me wonder why additional text is needed in supplementary material. Couldn't it be incorporated into the manuscript? The portion about the feature tracking seems to be important, but is only briefly mentioned in the main text.

We have added a few sentences in the main manuscript to describe feature tracking method, please see Line 16-18 on page 4: "We compared this new velocity map with velocities estimated by feature tracking (done with Open Source Computer Vision Library: https://opencv.org/, uncertainty is typically <1 m d$^{-1}$ for a pair of images separated by one day), which is independent of interferometry and does not require phase connection". We also used RMS of velocities by feature tracking of stationary points to quantify the uncertainty, see the dashed white arc in Fig. 3(a) and the supplementary material.

Some of the figures could also use work: 1. The green lines in Figure 2 are almost undetectable, and the red and green lines will be difficult for some readers.

We have replaced the green lines with red lines. In our revised manuscript and supplement, we have tried to avoid using red and green for similar markers in the same figure.

2. Figure 6 is also really difficult to read, and it may be misleading in that the tidal response appears to grow in both the upglacier and downglacier directions, but is minimal somewhere in between. I understand that this is at least partially due to flow direction not corresponding with line of sight, but that needs to be made more clear. You could indicate that the smallest response occurs where the flow is perpendicular to line of sight.

The revised caption and manuscript should clarify the information in this figure (now Fig. 4).

Figure 4 (a, c, e) are used to show periodic variations, not long-term velocities. Apart from effects due to angles between flow directions and radar LOS, the observed velocity themselves

can have smaller amplitudes in between the upstream glacier and downstream glacier. This is especially true near the floating-grounding transition zone, where tidal responses on horizontal (out of phase with tides) and vertical (in phase with tides) directions can diminish each other, making the amplitude smaller.

It is not necessarily true that the smallest response occurs where the flow is perpendicular to line of sight. As we show in later discussion (e.g., Fig. 9 in the revised manuscript), the long-term LOS velocity can be the smallest (~0) when the flow is perpendicular to LOS, however, amplitude of the periodic signal is not necessarily the smallest here.

To make this figure more clear, we have also removed pink lines (glacier maximum/minimum extents) on (b, d, f).

3. Figure 9: It would be nice to see the location of the radar on these maps. Its pretty obvious where its located in the MLI images in previous figures, but not in the DEMs.

We have added a red dot in our revised manuscript to show the radar location (now Fig. 7).

4. Figure 12: I'm not sure what purpose this figure really serves.

The figure was used to assess the forcing due to surface melting. Maximum in the K1 tide rate occurs ~ 6 h after local noon. Assuming there is a diurnal signal caused by surface melting which could peaks ~6 h after local noon, it would be superimposed on the observed velocities. When observed from a positive LOS direction, this would enhance the diurnal signal; when observed

from a negative LOS direction, this would diminish the diurnal signal. This could explain why the diurnal component of power spectral density plot in Fig. 2(C) is smaller compared to boxes A and B.

We have removed this figure, and simplified our explanation into a few sentences in the first paragraph in section 4.2: "As shown in the normalized PSD in Fig. 2(C), the diurnal constituent is less obvious compared to Fig. 2(A, B): Assuming speed maxima caused by surface melting lags local noon by 6 h, it will be in phase with the K1 ocean tide rate. Due to the geometry difference, TRI-observed LOS diurnal tidal signal will be superimposed on a negative (box C) or positive (box A and B) diurnal signal associated with surface melting, decreasing or enhancing the observed signal. Thus the diurnal constituent in Fig. 2(C) is smaller compared to the other two areas".

5. Figure 13: Its really difficult to see the arrows in panel a. And why are feature tracking velocities projected onto line of sight? That seems backward and misleading to me, as it gives the impression that velocity variations are toward/away from the radar. I think it would be better to project the LOS velocities into map view, and then talk about what causes the variations in "true" velocity.

The figure has now become Fig. 9. In our revised manuscript. We show the 2-D velocity map from feature tracking in Fig. 3(a), and also show 2-D velocities for the other two campaigns in the supplement. However, in this figure, we show how the 2-D velocity would look in terms of radar LOS in order to explain to non-radar specialists some of the subtleties inherent in this imaging technique.

We projected feature tracking velocities onto LOS because we are trying to show points with velocities that are perpendicular to the radar LOS. As shown in the figure, the transition zone between dark red arrows and dark blue arrows match well with white areas on the LOS velocity map, indicating real velocities that are perpendicular to the radar LOS direction. We could plot these arrows using real velocities, however, the arrows would then not directly linked to the phase derived LOS velocities. We therefor believe it is better to project velocities onto LOS for this figure, and readers who read the caption hopefully will not misunderstand the figure.

We have changed the arrow colors to dark red (towards the radar) and dark blue (away from the radar) to allow better visibility.

Some specific comments: Page 1, Line 5: I would not say that ice is locally thin if the freeboard is less than 125 m!

We have changed it to "much thinner compare to ice >1 km upstream".

Page 1, Line 22: "down dipping upstream bed" is confusing. Do you mean retrograde bed?

Yes and we have adopted the referee's suggestion using "retrograde".

Page 2, Line 2: "ice speed accelerates" – speed doesn't accelerate, but ice does

We have removed "speed".

Page 2, Line 15: "grounding line position" is not really a "basal condition"

This now becomes Line 14 on Page 2. We have rewritten this sentences to "Currently, it is challenging to observe grounding line position directly when it lies near the calving front".

Page 2, Line 30: "through a calving season"? This makes it sound like you were operating the TRI all summer long, which is misleading.

This now becomes Line 28-30 on Page 2. We have rewritten this sentence to "Here we use TRI measurements obtained in three summer campaigns, but at different stages (early versus late summer) of the calving season, to investigate tidal response and the evolving glacier front through Jakobshavn Isbræ's calving season".

Page 5, Line 19: "The amplitude" of? I'm not quite sure what this refers to.

This now becomes Line 21-22 on Page 5. We have changed this sentence to "The amplitude of variation is magnified by frequency". Please see Supplementary Text S3 for detail.

Pages 5-6 (and supplement): The step-change in ice melange thickness is interesting and suggests that the ice melange has a "terminus". However the discussion is highly speculative and doesn't really fit in this subsection, which is about tidal analysis. Also, I don't buy the idea that the change occurs because of some bedrock topographic feature. I wonder if instead you are seeing the remnants of the winter melange that hasn't yet lost cohesiveness.

We have removed this discussion about the melange. The thick melange is not likely to be the remnants of the winter melange since we see some large calving events from Landsat-8 and Sentinel-2 satellite images earlier in the spring. Those calving events should have removed pro-glacial melange accumulated in winter, although re-freezing is possible. Another possibility is the instead of bedrock obstructions, the obstructions are on the sides of the fjord, ie the fjord near this point and occasionally becomes choked with ice.

Page 7, Lines 19-20: This seems like a pretty big assumption, considering that other studies have suggested year-to-year variability in tidal response.

This has now become the first paragraph on Page 7. We acknowledge that year-to-year variability in tidal response is possible. However, position time series of the calving front derived from satellite images supports this assumption. As shown in Fig. 5, Jakobshavn Isbræ had a relatively regular advance and retreat over the five year observation period. Joughin et al. (2008, 2014) documented strong seasonality in speed that shows a good inverse correlation with the seasonally varying length of a short ice tongue. In addition, our 2015 and 2016 data provide a comparison of tidal responses both in early summer but from different years, they have similar behavior. There is also similarity between our 2012 campaign and Podrasky et al (2014) in late summer but in different years. For these reasons, we assume a regular variation in tidal response over the 5 five year spanning our observation period.

We have added a few more sentences to the first paragraph on Page 7 to support this assumption.

Page 8, equation 3: Double check this equation. I'm pretty sure that V_los and dh/dt should be swapped.

We apologize for the confusion, we are grateful that the referee pointed out some questions on this equation. $H_0$ is the mean height difference between the radar and the target, not "the mean height of the target" as we wrote in our previous draft. And $V_{los}$ represents the TRI-observed LOS velocity (not "vertical" as in our previous draft). We have corrected this. Apart from that, the equation remains unchanged. The figure below shows these relations, where blue dot represents the radar, red square shows the mean vertical position of the target, red dot is the vertical position at current time, other parameters are the same as described in the manuscript. We have added this figure to the supplement.

[Figure]

Page 8, Line 17: dh/dt ~ 0.1*tidal rate in the melange due to buoyancy effects, and less than that for the glacier.

This is in Line 7 on Page 8 now. Please see our response to the previous comment. As dh/dt denotes vertical component of ice velocity, it is approximately equal to tide rate in the melange. We have added the phrase "and less than that for the glacier".

[revised manuscript text omitted]

**Contents:**

Text S1: Method to fix phase offsets in the 2012 data

Text S2: PSD plots for 2012 and 2015

Text S3: Tide and tidal rate

Text S4: Phase lag maps for K1 and S2 tidal frequencies

Text S5: Geometry between velocity and radar LOS

Text S6: Movies shown major calving (-like) events during 3 campaigns

Figure and movie contents are introduced in the text.

**1. Method to fix phase offsets in the 2012 data**

Ice velocity estimates have some significant offsets in the 2012 data. These offsets represent phase discontinuities, clustering at integer multiples of the radar wavelength (Fig S1(a)). They occur because the 2012 data were acquired with a scan rate that was slow compared to ice velocity, such that ice could move more than one radar wavelength relative to adjacent area during the 3 minute scan interval (data in later years were acquired with a faster scan rate). Compared to fast moving ice, rocks near the TRI can be treated as stationary objects. We used a rock reference point ~0.2 km away from the radar as a reference for phase unwrapping. To correct the phase discontinuities caused by phase unwrapping error, we firstly used a model to fix the phase offsets for a single point on ice (P0, yellow point in Fig S1(b)), then used it as reference point for phase unwrapping to get velocity maps relative to this point. Ice velocities relative to the stationary rock were then generated by adding the modeled velocities of P0 to those relative velocities. We selected data obtained between 6 August and 10 August for the following process because there were continuous measurements during this period and only one small calving event. Here are the 4 steps used to model the velocity of P0:

1) Use the same data processing procedure as described in *Voytenko et al.* [2015)] and *Xie et al.* [2016], generating time series for all velocity maps. The generated velocities are absolute velocities, plus/minus some value equal to an integer number of wavelength jumps. The TRI instrument transmits Ku-band microwaves with wavelength of 1.74 cm. In a 2-way measurement system, 1 wavelength jump in unwrapped phase leads to 0.87 cm change in the LOS displacement, equal to 4.2 m d$^{-1}$ offset in LOS velocity when the repeating time of measurement is 3 minutes. From Fig S1(a) we infer that most of the unwrapped phases for P0 have 4 cycles of phase offset.

2) Remove "apparent outliers" by using the modified Z-score method [*Iglewicz and Hoaglin, 1993*]. For the *i*th observation $x_i$, its Z-score is:

$$Z_i = 0.6745(x_i - \tilde{x}) / \mathrm{MAD} \tag{S1}$$

where MAD denotes the median absolute deviation, and $\tilde{x}$ is the median value. Observations with absolute value of the modified Z-scores >3.5 were considered as outliers and removed. Then we subtracted a 2nd-order polynomial curve to remove the possible response to calving events. The method by *Davis et al.* [2014] was then used to estimate the periodic components caused by tidal variations. We chose 3 sinusoids with the same frequencies as the K1/M2/S2 tidal constituents (see the main paper for more information about the tidal constituents at Jakobshavn Isbræ). The solid blue curve in Fig S1(c) shows the best fit of a 2nd-order polynomial + 3 sinusoids to the observed time series. The dashed blue curves mark 3 times the value of the root-mean-square (RMS) of the residuals.

3) Shift all observations (including "apparent outliers" in step 2) upwards/downwards by 4.2×N m d$^{-1}$, where N = 0, ±1, ±2, ±3, ±4, ±5, etc. Observations who's shifted values do not fall into the 3×RMS space defined in step 2 are labeled as outliers (grey dots in Fig S1(c)) and removed. Values that fall within the 3×RMS space are then shifted by 4 cycles of phase offset (16.8 m d$^{-1}$) to eliminate the jumps derived by feature tracking method in

step 1.

4) Apply a median filter (kernel size = 3) to the time series from step 3, and then use the same model in step 2 to fit the new time series. Estimated parameters are then used for further processing. Fig S1(d) shows the least square fit of the final fixed time series for P0.

For all other points on the TRI image, their velocities are estimated by adding the modeled velocity of P0 to their relative velocities. Fig S2 shows phase offsets-fixed LOS velocities for selected points. Black dots are velocity estimates when using a stationary rock point as the reference point for phase unwrapping (shifted upwards by 4 cycles of phase offsets). From Fig S2 we see that phase jumps are greatly reduced by this method, especially on the glacier. In the mélange, time series shortly after two calving events (on 5 August and 9 August, see blue arrows in Fig S1(a)) still have some jumps, mainly due to phase breaks caused by rapid ice motion after calving events. In the tidal analysis section of the main paper, we omit data for the mélange acquired near these 2 calving events.

We used feature tracking (done with OpenCV: http://opencv.org/) as an independent method to examine the phase offset "fixed" velocities from interferometry. And used stationary or near-stationary points near the radar as references to calculate uncertainties in feature tracking, which is <1 m d$^{-1}$ in both x and y components for velocity estimates based on a TRI intensity image pair separated by 1 day. Fig S3(a) is the median LOS velocity map from a 1 day sequential measurements in the 2012 campaign. Fig S3(b) shows a velocity map (projected onto the LOS direction) derived by feature tracking using two TRI intensity images acquired at the beginning and end of the day (red means moving towards the radar, yellow means moving away). Fig S3(c) is the difference between (a) and (b). Red means the velocity estimated by interferometry is larger than the velocity estimated by feature tracking, yellow means it is less. For most points, the difference is much smaller than 1 cycle of phase jump (4.2 m d$^{-1}$). On the north edge of the mélange, there is a small area where offsets still exist, presumably caused by phase breaks due to discontinuities in the TRI maps. One way to solve this type of phase jump is to use shorter repeat time when collecting data. However, this problem won't affect the following tidal analysis if the sign of LOS velocity has not been changed by these phase jumps. Because this study focuses on short time-scale tidal responses, the large background velocity was detrended before tidal analysis. We also noticed that the velocity differences near the radar are much larger, up to a few cycles of phase jumps. This is due to the fact that the phase data for stationary areas near the radar have no discontinuity problems in speed as on the ice. The method described above has thus introduced some artificial jumps in the stationary near field.

Fig S4 and S5 show comparisons of velocity estimates by interferometry and feature tracking for 2015 and 2016. There are no systematic differences for these two years. Except for some random errors, significant differences only appear in isolated patches where the phase map lacks continuity, or in places close to where calving events have occurred. Thus, we did not apply the same method used for 2012 to the 2015 and 2016 data. Instead, we did phase unwrapping by using stationary points on rocks only.

**2. PSD plots for 2012 and 2015**

Fig S6, power spectral density (PSD) for selected areas in 2012
Fig S7, power spectral density (PSD) for selected areas in 2015

**3. Tide and tidal rate**

Fig S8 shows predicted and observed tide and tidal rate during 2015 campaign. In this study, tidal rate is defined as the 1st time derivative of tidal height. For a tide signal:

$$H(t) = A \cos(2\pi ft + \varphi) \tag{S2}$$

where $A$ is the amplitude, $f$ is the frequency, $\varphi$ is the phase. The tidal rate is:

$$H'(t) = 2\pi fA \cos(2\pi ft + \varphi + 0.5\pi) \tag{S3}$$

Compare to tidal height, the amplitude of tidal rate has been amplified by $2\pi f$. Due to differentiation, the phase difference between ice velocity and tidal rate is the same as the phase difference between ice position and tidal height.

**4. Phase lag maps for K1 and S2 tidal frequencies**

Fig S9, phase lag in time (h) for K1 and S2 tidal frequencies.

**5. Geometry between velocity and radar LOS**

Fig S10, geometry used to project velocity onto radar LOS direction.

**6. Movies shown major calving event during 3 campaigns**

Mov S1, Major calving events in 2012.
Mov S2, Major calving events in 2015.
Mov S3, Major calving (-like) events in 2016.

[Figure]

**Figure S1. (a)**, LOS velocity time series (black dots) of P0 on (b), when using a stationary point on rock as reference for phase unwrapping. Pink bars show velocities by feature tracking method (shifted downwards by phase jumps of 4 cycles of radar wavelength), separated by ~1 d. Blue arrows mark visible calving events on TRI intensity images, two light blue arrows represent calving events that were in observation gaps. **(b)**, Point (P0) chosen as reference point for phase unwrapping to estimate relative velocities. Red arrows show 2-D velocity map by tracking two TRI intensity images between 00:01:00 7 August and 00:00:00 8 August. Background is a TRI intensity image acquired on 6 August 2012. Arrows within the dashed white arc are used to assess the uncertainty of feature tracking, the RMS is <1 m d$^{-1}$. **(c)**, LOS velocity time series between 6 August and 10 August (dots between two grey lines in (a)). Solid blue curve shows the best fit after "apparent outliers" (defined by modified Z-score method) removed, by using a 2nd-order polynomial + 3 pairs of sinusoids model. Dashed blue curves show 3 times RMS space of the residuals. Solid and dashed Light blue curves are blue curves moved upwards/downwards by integer numbers of radar wavelength cycle. Dots fall into the 3×RMS spaces (red) were used in (d), grey dots were removed as outliers. **(d)**, Time series from (c) after shifting upwards by phase jump of 4 cycles of radar wavelength, and applying a median filter with a window-size of 3 (equals to 9 min in this case, where the repeating time of measurement is 3 min). Upper blue curve shows the best fitting by the same model as (c), but used filtered time series. Lower blue curves show the 2nd-order polynomial and sinusoids components, and are offset for clarity. Cyan curve shows local tidal height rate prediction.

[Figure]

**Figure S2.** LOS velocities for selected points, after phase jumps fixed. (**a**), Point locations for P1 - P8. (**P1-P8**), Red dots are phase jumps fixed time series, black dots are time series when choosing a stationary point as reference for phase unwrapping, and were shifted upwards by 4 cycles of phase jumps.

[Figure]

**Figure S3.** Comparison between velocity estimated by interferometry and feature tracking. (**a**), Median LOS velocity map by interferometry method for 7 August 2012, after phase jumps fixed. Background is a Landsat-7 image acquired on 6 August 2012, white stripes are data gaps. (**b**), Velocity map by feature tracking method, projected onto radar LOS direction. Red moves towards the radar, yellow away. (**c**), Difference between (a) and (b), red when velocity by interferometry is larger than by feature tracking, yellow vice versa. 4.2 m d⁻¹ equals to 1 cycle of phase jump.

[Figure]

**Figure S4.** Comparison between velocity estimated by interferometry and feature tracking in 2015. (**a**), LOS velocity map by interferometry. (**b**), 2-D velocity map by feature tracking. Arrows within the dashed white arc are used to assess the uncertainty of feature tracking, the RMS is <1 m d$^{-1}$. (**c**), Velocity map by feature tracking method, projected onto radar LOS direction. Red moves towards the radar, yellow away. (**d**), Difference between (a) and (c), red when velocity by interferometry is larger than by feature tracking, yellow vice versa. 8.4 m d$^{-1}$ equals to 1 cycle of phase jump, when repeating time was 1.5 min.

[Figure]

**Figure S5.** Comparison between velocity estimated by interferometry and feature tracking in 2016. Color and arrow denote the same as Fig. S3. In (c), 6.3 m d$^{-1}$ equals to 1 cycle of phase jump, when repeating time was 2 min. The 2-D velocity map is shown in Fig. 3(a) of the main paper.

[Figure]

**Figure S6.** Stacked power spectral density (PSD) estimates of the LOS velocity time series for selected areas in 2012. Two 0.5 km × 0.5 km boxes (A and B) mark the selected areas. For PSD plots, each black line represents 1 pixel (10 m × 10 m) in the corresponding box. Yellow lines show the mean value. PSDs shown here are normalized. PSD analysis was done by using the Lomb-Scargle method (Lomb, 1976; Scargle, 1982). Red lines mark the frequencies of K1, M2, and S2 tide constituents.

[Figure]

**Figure S7.** PSD plot for selected samples in 2015. Lines and colors denote the same as in Fig. S6.

[Figure]

**Figure S8.** Predicted and observed tide and tidal rate during 2015 campaign. Tide is the dashed line, tidal rate is in solid line. Cyan color represents predicted tide and tidal rate, based on Richter et al. [2011]. Red color represents observed tide and derived tidal rate from a mooring at the mouth of the fjord.

[Figure]

**Figure S9.** Phase lag maps of signals with K1 and S2 frequencies for each campaign. (a-b) for 2012, (c-d) for 2015, (e-f) for 2016. (a, c, e) show phase lags in time (h) for K1 frequency signal. (b, d, f) for S2 frequency signal. areas where SNR<1.5 are omitted.

[Figure]

**Figure S10.** Geometry used to project velocity onto radar LOS direction. Blue dot represents the radar, red square shows the mean vertical position of the target, red dot is the vertical position at current time. $H_0$ is the mean height difference between the radar and the target. $L$ is the horizontal distance between the radar and the target. $h$ is the vertical movement relative to $H_0$. $dh/dt$ is the vertical component of ice velocity. $V_{los}$ represents the TRI-observed LOS velocity. $V_{perp}$ represents ice velocity projected onto perpendicular direction of the radar LOS.